# Climate-driven tradeoffs between landscape connectivity and the maintenance of the coastal carbon sink

Kendall Valentine ●[1,2] ✉, Ellen R. Herbert[1,3], David C. Walters[1,4], Yaping Chen ●[1], Alexander J. Smith ●[1] & Matthew L. Kirwan ●[1]

Ecosystem connectivity tends to increase the resilience and function of ecosystems responding to stressors. Coastal ecosystems sequester disproportionately large amounts of carbon, but rapid exchange of water, nutrients, and sediment makes them vulnerable to sea level rise and coastal erosion. Individual components of the coastal landscape (i.e., marsh, forest, bay) have contrasting responses to sea level rise, making it difficult to forecast the response of the integrated coastal carbon sink. Here we couple a spatially-explicit geomorphic model with a point-based carbon accumulation model, and show that landscape connectivity, in-situ carbon accumulation rates, and the size of the landscape-scale coastal carbon stock all peak at intermediate sea level rise rates despite divergent responses of individual components. Progressive loss of forest biomass under increasing sea level rise leads to a shift from a system dominated by forest biomass carbon towards one dominated by marsh soil carbon that is maintained by substantial recycling of organic carbon between marshes and bays. These results suggest that climate change strengthens connectivity between adjacent coastal ecosystems, but with tradeoffs that include a shift towards more labile carbon, smaller marsh and forest extents, and the accumulation of carbon in portions of the landscape more vulnerable to sea level rise and erosion.

Coastal ecosystems sequester disproportionate amounts of soil carbon compared to terrestrial ecosystems[1], making them targets for potential climate change mitigation by land managers and policy makers. Sea level rise (SLR) both facilitates and threatens coastal carbon accumulation. For example, coastal marshes have a potential negative carbon-climate feedback, where soil carbon accumulation rates (CAR) at a given point on the marsh surface increase in response to SLR[2,3]. However, SLR also threatens the extent of coastal habitats[4–6], driving accelerated coastal forest mortality via saltwater intrusion[7–9] and marsh drowning[10–12]. Therefore, it is unclear if a negative carbon-climate feedback will persist, as the fate of coastal carbon depends not only on how CAR responds to SLR, but also on how the size, configuration, and interactions of the coastal system respond.

Our understanding of coastal blue carbon is largely based on the discretization of the coast into distinct habitats (e.g., marshes, mangroves, seagrass) with static boundaries to determine carbon stocks and CAR[13–15]. Individual systems have different and even opposite responses to SLR. For example, marsh soil CAR is expected to increase with SLR[2], but coastal forest biomass is expected to decrease with SLR-induced saltwater intrusion[16,17]. Given these contrasting responses, the net impacts of SLR on coastwide carbon remain largely unknown and is not explicitly included in models of coastal carbon dynamics. Exchange, or connectivity, of carbon between adjacent ecosystems is

[1]Virginia Institute of Marine Science, College of William and Mary, Gloucester Point, VA, USA. [2]School of Oceanography, University of Washington, Seattle, WA, USA. [3]Ducks Unlimited, Memphis, TN, USA. [4]U.S. Geological Survey Eastern Ecological Science Center, Laurel, MD, USA. ✉e-mail: kvalent@uw.edu

substantial[18,19] but poorly quantified and is typically excluded from carbon budgets despite being a requisite of blue carbon accounting protocols[20]. For instance, sediments eroded from the marsh edge redeposit on the marsh surface[21,22], and undoubtedly lead to the recycling of carbon between marshes and mudflats[22]. Quantifying the source of coastal carbon is important to evaluate its impact on climate feedbacks. To reduce elevated atmospheric carbon, accumulated carbon must be derived from newly-fixed carbon, as carbon redistributed from erosion does not remove atmospheric carbon.

Ecosystem connectivity is critical in regulating ecosystem functions[23] and has been demonstrated to lessen the effects of stressors[24] in a variety of ecosystems[25,26]. Although the net response of the coastal landscape to changes in SLR will likely depend on connectivity between ecosystems, Earth system models fail to address the complex exchange of sediment, water, and nutrients between terrestrial and oceanic systems[27]. Within the geomorphic context, connectivity describes the material transfer between components of the landscape (i.e., structural configuration)[28]. Here we extend this concept to address connectivity, or material transfer, of carbon across the entire coastal landscape. Specifically, we investigate how dynamic structural configuration of the coast affects the transfer of carbon between adjacent ecosystems and how this impacts overall ecosystem

function. We present an exploratory landscape-scale model (CoLT, Coastal Landscape Transect model) that connects three distinct coastal ecosystems (bay-marsh-forest) and show that SLR enhances the connectivity of carbon, resulting in an increase in the magnitude of the coastal carbon sink up to a tipping point, after which elevated SLR and connectivity decreases carbon stocks and ecosystem function.

## Results and discussion
### Model approach and basic behavior
Our modeling approach simulates coastal landscape evolution and carbon transport along a transect connecting three distinct coastal habitats: bays, marshes, and upland forests (Fig. 1; "Methods"). Previous work typically models the geomorphology or carbon storage processes of an individual system[29,30]. In cases where multiple systems are considered, either the geomorphology[31,32] or carbon storage[33] is modeled, but not both. Here we coupled multiple habitats and simultaneously modeled geomorphic and carbon cycling of a connected system and explored the outcomes in an idealized coastal transect. This type of exploratory model has been used for decades in geomorphology as a way to identify and understand feedbacks in nonlinear systems[34,35]. We started with a geomorphic transect model that considers the key drivers of topographic evolution of bay

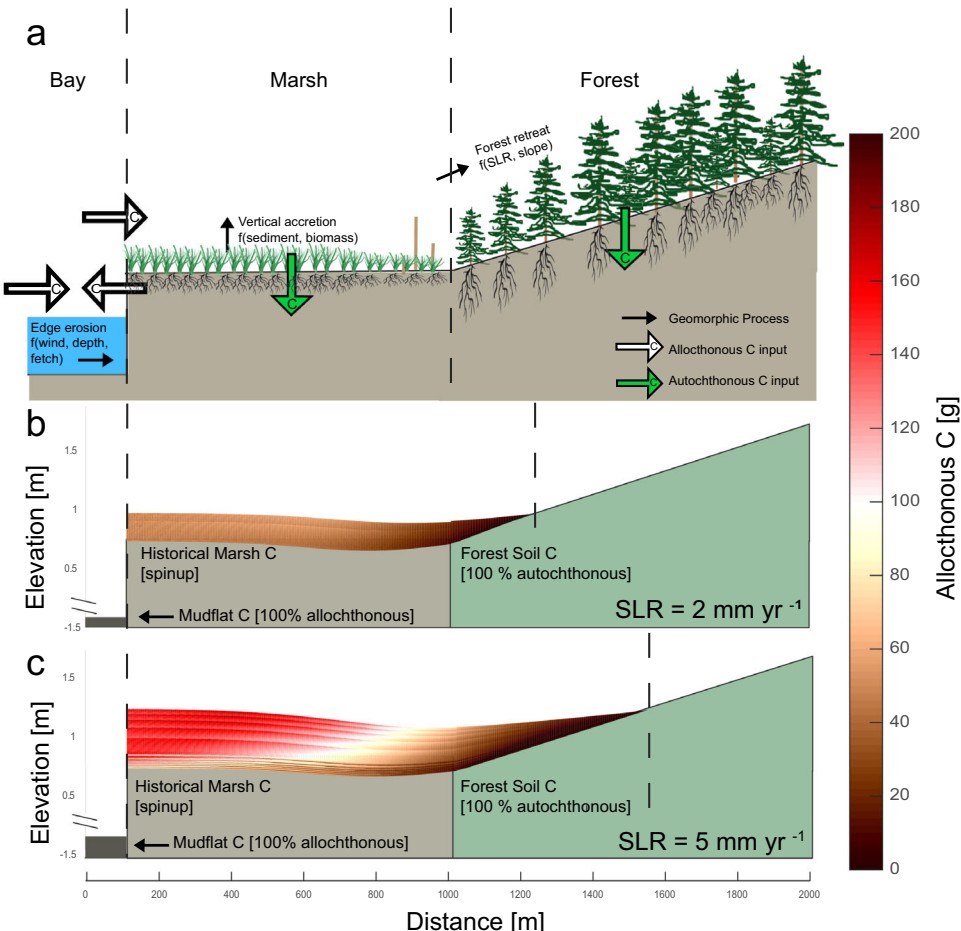

**Fig. 1 | Bay-Marsh-Forest transect demonstrates the exchange of carbon between ecosystems and the accumulation of allochthonous carbon in marshes.** Schematic of 2D transect model of the bay-marsh-forest system representing all modeled processes (**a**). Geomorphic processes are indicated with black arrows, while carbon processes are in green (autochthonous, Supplementary Fig. 4) and white (allochthonous, shown in panels **b** and **c**). The coastal transect was subjected to low [2 mm yr⁻¹] (**b**) and moderate [5 mm yr⁻¹] (**c**) rates of sea level rise (SLR), which resulted in more allochthonous carbon (C) under high rates of SLR. Model

experiments were conducted under a 50 mg L⁻¹ sediment supply and a 1.4 m tidal range. Color shadings along scale on right indicate the amount of allochthonous carbon [g], with red representing higher carbon content. Underlying stratigraphy was generated during the model spinup. x-axis distance is relative to initial shoreline position and y-axis is relative to initial sea level. Vertical dashed lines delineate bay-marsh and marsh-forest boundary positions at the end of the model simulations. Total carbon is presented in Supplementary Fig. 5.

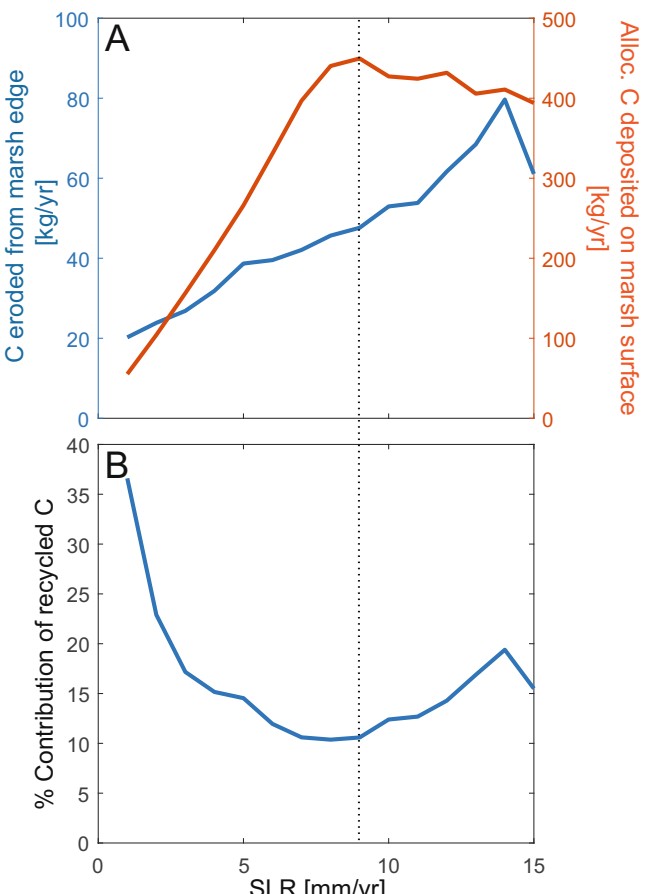

**Fig. 2 | Exchange of carbon at the marsh-bay interface increases with sea level rise (SLR) rate, and recycled carbon remains important in all SLR scenarios. A** Fluxes (kg yr⁻¹ per meter of marsh edge) of carbon eroded from the marsh edge into the bay (blue line) and of allochthonous carbon (C) deposited on the marsh platform from the bay (red line). The carbon flux from the marsh to the bay represents the mass flux of carbon eroded from the marsh edge, averaged over the last 50 years of the model experiment. The carbon flux from the bay to the marsh represents the mass flux of allochthonous carbon deposited on the marsh surface (surface deposition), averaged over the last 50 years of the model experiment. **B** The material eroded from the marsh edge makes up ~15% of the total carbon deposited on the marsh surface. This contribution decreases until an intermediate rate of SLR, and then increases as SLR increases. Colored envelopes show the variation (minimum and maximum) in results with suspended sediment concentration (SSC) = 20–60 mg L⁻¹.

bottoms, marshes, and forests[31]. The bay width and depth evolve as a function of wind speed, fetch, and water depth[36]. Marshes accrete as a function of mineral sediment deposition and biomass production[29]. Forests migrate passively as a function of upland slope and SLR[31]. Sea level is calculated yearly and used to determine marsh inundation, and therefore mineral and organic sediment deposition, as well as plant biomass and therefore carbon production, each year. We do not explicitly model ponds or channel development. Sensitivity analysis demonstrates that the conceptual insights hold across a wide range of suspended sediment concentrations and tidal amplitudes (Supplementary Figs. 3–5).

We then modeled marsh carbon accumulation following a soil-cohort approach in which organic matter evolves dynamically through time according to the balance between belowground organic matter production and decomposition, and their dependence depth within the soil profile[29,30]. While previous soil cohort models explore these processes at a single point on the marsh surface, here we focus on spatial gradients across the marsh surface, connectivity between

adjacent coastal ecosystems, and the exchange of carbon between marshes and bays. Consequently, we distinguish between carbon produced and retained on the marsh surface (autochthonous carbon) from carbon that is exchanged between the bay and marsh (allochthonous carbon). Autochthonous carbon is modeled as a function of belowground biomass, and it decomposes at a depth-dependent rate that goes to zero once the material is buried below the active rooting zone (0.4 m below the marsh surface)[37]. Belowground biomass, represented as a monoculture of *Spartina alterniflora*, is modeled as a parabolic function of elevation which peaks at an intermediate elevation ($B_{max} = 1000 \, \mathrm{g \, m^{-2}}$)[31,36,38]. Allochthonous carbon is sourced from the bay and decreases with distance from the marsh edge. In this model the bay bottom is unvegetated, and therefore all carbon in this ecosystem is considered allochthonous. While some shallow bays are vegetated with seagrass[39], they induce further geomorphic feedbacks. For example, seagrass colonization depends on the elevation of the bay bottom; however, this model represents the bay bottom with a single elevation. Allochthonous carbon flux to the bay is sourced from the marsh edge. The eroded carbon is allowed to deposit and contribute to the bay bottom soil carbon whenever the geomorphic conditions allow for bay bottom accretion[36]. For simplicity, the model assumes that allochthonous carbon is recalcitrant and does not decompose. Although some studies report the decomposition of allochthonous carbon following disturbance[40], allochthonous carbon can be millennia old[22,41] and dominantly comprised of unreactive carbon[41]. This assumption potentially overestimates the amount of organic matter in the system, as marsh erosion exposes and disturbs previously-buried carbon. However, marsh edge erosion makes up a small component of total allochthonous carbon (15%, Fig. 2B); the remaining allochthonous carbon is from the external sediment supply and resuspension of the bay bottom. These sources are repeatedly disturbed and any remaining carbon is recalcitrant and tightly bound to sediment[41]. Although both autochthonous carbon production and decomposition will be altered with a changing climate[42,43], the non-linear interactions between temperature, $CO_2$, and nutrients are complex and beyond the scope of this model. Therefore, we do not capture all climate-carbon feedbacks but focus solely on the climate effect of SLR. Furthermore, we limit carbon processes to particulate carbon that is directly associated with sediment transport processes and biomass production. The outwelling of carbon, in both dissolved and particulate form, from groundwater, is an important part of the carbon budget in the coastal zone[44] and for plant productivity[45], but is not resolved here.

We additionally developed a new module for biomass and soil carbon accumulation in mature coastal forests. In the model, forest biomass is modeled as a logarithmic function of elevation. Field and remote-sensing observations suggest that coastal forest biomass is lowest at the marsh-forest transition due to intensified seawater intrusion[17,46]. Tree biomass gradually increases with elevation up to a point where trees are no longer stressed by encroaching seawater. Soil CAR in coastal forests is comparatively low[47] and is thought to be near a carbon saturation value[48]. However, at the marsh-forest transition, elevated inundation increases the amount of organic matter preserved and stored in the soils[17]. Therefore, the model we developed simulates coastal forest soil carbon as an exponential decay function of the elevation relative to sea level. Forest soil CAR is the highest at the marsh-forest transition and declines with increasing elevation to a constant low baseline value that represents the slow CAR in terrestrial forest soils. We assume a mature forest (forest age >80 years), where the balance between carbon deposition and decomposition is in steady state[49]. Therefore, the model considers a single value for net carbon accumulation, based on field measurements[17], that reflects both deposition and decomposition within a timestep, rather than separately modeling productivity and decomposition in live forest soils. However, all soil carbon in the forest is considered

autochthonous, and therefore becomes subject to additional decomposition when it becomes overlain by marsh.

In an initial set of experiments designed to understand basic model behavior, we subjected the model to low [2 mm yr⁻¹] and moderate [5 mm yr⁻¹] SLR scenarios. Initial organic soil layers were generated with a 550-year model spin-up under a constant rate of SLR [1 mm yr⁻¹], in which marsh and bay bottom elevations equilibrated to a low rate of SLR that reflects pre-industrial conditions. Following the spin-up period, the transect was subjected to a new, faster rate of SLR for 100 years, a duration that allows the system to approach equilibrium. The results illustrate that our model (Fig. 1b, c, Supplementary Fig. 1) accurately captures key processes observed in the field in response to accelerated SLR, including enhanced marsh productivity[50], increased vertical accretion rates[51], increased in-situ CAR[2,3], and the landward migration of marshes and organic rich soils[52,53]. As expected, the marsh sediment profile is deeper and the marsh platform is wider with moderate SLR (Fig. 1c) compared to historical SLR (Fig. 1b), reflecting increased vertical accretion rates and faster marsh migration. Correspondingly, CAR (Fig. 1b, c, Supplementary Fig. 2) and marsh productivity (Supplementary Fig. 1) were also higher with moderate SLR than with historical SLR.

Aside from capturing important processes known to influence coastal carbon cycling, our model also provides new insights into the effects of dynamic changes in ecosystem size on coastal carbon stocks, and the relative importance of organic matter exchange between individual components of the coastal landscape. For example, the model demonstrates that coastal landscapes change in size and position through time (Fig. 1), associated with the erosion of the bay bottom[32,54] and marsh edge[32,54] and the migration of marsh into retreating forests[12,55], both of which reduce the coastal carbon sink[16,17,56]. Forest retreat rates increased with SLR, leading to a smaller forest extent and a larger marsh extent in high SLR scenarios compared to low SLR scenarios (Fig. 1).

We also observe that the mass of allochthonous carbon is highest near the marsh edge in all scenarios and that autochthonous carbon dominates the marsh interior (Fig. 1b, c). Across the marsh platform, 2.3 times more allochthonous carbon is stored under moderate rates of SLR [5 mm yr⁻¹, 13 Mg C] compared to the slow rate of SLR [2 mm yr⁻¹, 5.6 Mg C]. Within 100 m of the marsh edge, where allochthonous carbon inputs are more important[57], 2.6 times more allochthonous carbon is stored under moderate rates of SLR compared to slow rates. The increased accumulation of allochthonous carbon is driven by greater accommodation space and thicker marsh soils[3,29], but also represents the increased import of carbon from the bay to the marsh.

## Connectivity of carbon between marshes and bays increases with SLR

Connectivity between adjacent ecosystems tends to increase ecosystem stability[58,59]. The exchange of sediment and nutrients between bays and marshes is important for the long-term resilience of both systems[32,60]. To explore the impacts of connectivity on coastal carbon, we conducted a second set of experiments under a larger range of SLR rates that represent the range of potential SLR rates in the next century [1–15 mm yr⁻¹]. We quantified the amount of carbon exchange between the bay and the marsh (i.e., connectivity) during two processes: carbon released into the bay as the marsh edge erodes (carbon moving from marsh to bay) and carbon deposited on the marsh surface during inundation (carbon moving from bay to marsh). The efflux of carbon from the marsh due to edge erosion increases with SLR rate (Fig. 2A). The carbon flux from marsh edge erosion increased [10 vs. 16 kg yr⁻¹] when rates of SLR increased from 2 mm yr⁻¹ to 5 mm yr⁻¹ (Fig. 2). Marsh elevation increased similarly for this same change in SLR rate [-20 vs -50 cm over 100 years near the marsh edge], representing vertical accretion rates that keep pace with SLR (Fig. 1). Given that the rate of

edge erosion remains nearly constant regardless of SLR [-1 m yr⁻¹], this finding uniquely suggests that increased allochthonous carbon exchange results from both the larger marsh elevation relative to the bay bottom (i.e., the height of the eroding scarp) and the larger carbon stocks in marsh soils developed under higher SLR rates. Both processes result in more carbon-dense material being eroded from the marsh edge at higher rates of SLR, and therefore greater exchange of carbon across the marsh-bay boundary. Likewise, the allochthonous carbon deposition on the marsh platform from inundation increases with increasing SLR (Fig. 2A). However, the effect of increased connectivity has diminishing returns at extreme rates of SLR [>8 mm yr⁻¹]. At high rates of SLR, increased accommodation space leads to more deposition on the bay bottom[32], thus decreasing the amount of sediment and carbon remaining in the water column to be delivered back to the marsh platform. At this point, the amount of recycled carbon (i.e., carbon eroded from the marsh and then redeposited on the marsh) becomes increasingly important for deposition on the marsh surface (Fig. 2B), but the amount of material eroded from the marsh edge cannot sustain long-term marsh growth. Together, these experiments illustrate that coastal carbon cycling is strongly influenced by complex feedbacks between marshes and mudflats that would be difficult to foresee with field observations or with numerical models of individual ecosystems.

Previous work has identified a strong linkage between SLR and CAR for a given point on the marsh platform alone. CAR has been observed to increase in parallel with historical SLR acceleration[61,62] and is highest in regions with rapid SLR[3,30,63]. Several hypotheses have been put forth to explain the observed pattern, including expanded accommodation space and the associated decreases in carbon saturation effects[3,29,30,63], enhanced organic matter production due to a vegetation shift towards more flood-tolerant species[30,61], and increased recycling of carbon from eroding marshes[2]. Consistent with previous work, we find a positive relationship between CAR and SLR at intermediate rates of SLR [1–11 mm yr⁻¹], as illustrated by the increase of CAR from 30 g m⁻² yr⁻¹ [1 mm yr⁻¹] to 180 g m⁻² yr⁻¹ [10 mm yr⁻¹] (Fig. 3). However, CAR decelerates with additional increase of SLR rates [>10 mm yr⁻¹] that induce widespread marsh drowning [e.g., CAR of 130 g m⁻² yr⁻¹ at SLR of 15 mm yr⁻¹] (Fig. 3). Autochthonous CAR increases with intermediate SLR rates in the model simulations, as would be expected with more productive vegetation and expanding soil volumes that diminish decomposition effects[29,50].

Our model simulations offer two important insights. First, we identify the limit of the positive effects of SLR on CAR (Fig. 3). Though conceptually intuitive, this finding suggests that relationships between historical CAR and SLR cannot be projected indefinitely into the future[64]. Second, our model simulations offer a mechanistic interpretation for the relationship between CAR and SLR, highlighting the significance of carbon recycling. For example, our model simulations indicate that allochthonous sources can contribute up to 60 g C m⁻² yr⁻¹ to marsh soils when averaged across the entire marsh surface, making up to half of total marsh soil organic carbon on average [37–75% of CAR], and that deposition of allochthonous carbon increases with the rate of SLR and the duration of flooding (Fig. 3). Interestingly, both allochthonous and autochthonous CAR decrease at excessive rates of SLR that trigger marsh drowning and marsh width decline [>10 mm yr⁻¹]. However, total CAR remains higher under these elevated rates of SLR [>10 mm yr⁻¹] than for low rates of SLR [<5 mm yr⁻¹], driven in part by sustained allochthonous carbon deposition onto a progressively smaller marsh platform (Figs. 2, 3). Although CAR remains elevated, the decreasing trend in CAR with SLR (Fig. 3) corresponds to the decrease in connectivity (i.e., delivery of allochthonous carbon) (Fig. 2), supporting the importance of connectivity in marsh resilience. This general relationship holds for a range of external sediment supplies (Supplementary Figs. 3–5).

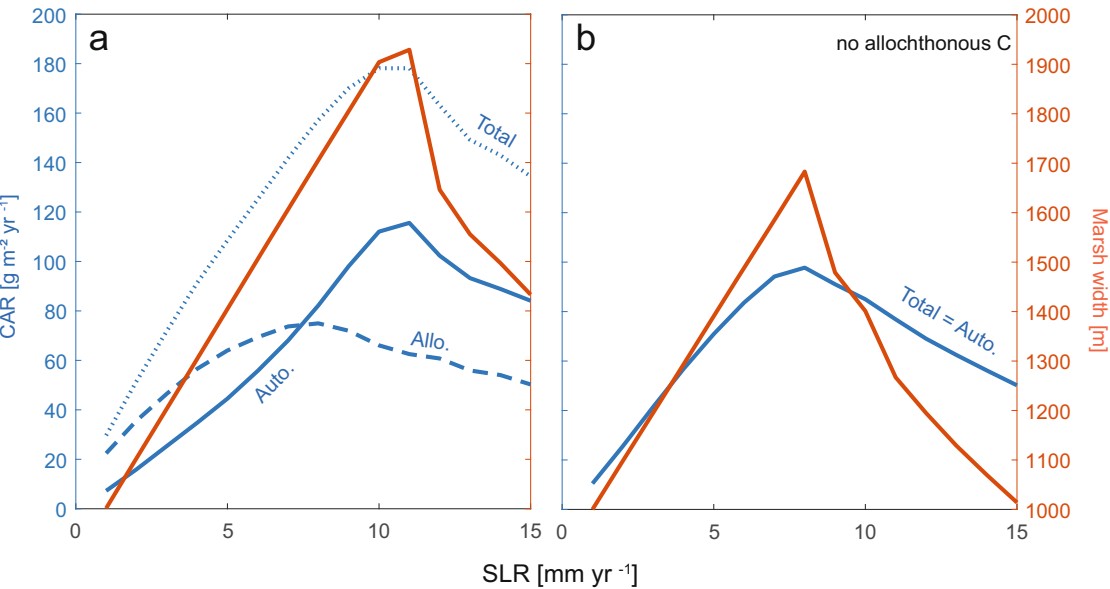

**Fig. 3 | Carbon accumulation and marsh width peak at intermediate rates of sea level rise (SLR). a** Carbon accumulation rate (CAR, blue lines, averaged over past 100 years over marsh platform) and marsh width at the end of the model simulation (red line). Dashed blue line indicates allochthonous (allo.) carbon, solid blue line indicates autochthonous (auto.) carbon, and dotted blue line indicates total carbon. While CAR decreases at extreme rates of SLR, it remains greater than the CAR at low rates of SLR. **b** CAR (blue) and marsh width (red) in model simulations where no allochthonous carbon was allowed to deposit (therefore total CAR is equal to autochthonous CAR).

Previous studies based on historical SLR rates [3–4 mm yr⁻¹] also reveal that the recycling of carbon from marsh erosion is an important carbon source to marsh soils[22,41,65]. Our finding that half of the carbon accumulated on the marsh is allochthonous demonstrates that recycling of carbon is important and suggests that under the current paradigm a substantial portion of carbon in a marsh does not contribute to climate mitigation. However, given recent advances in our understanding of carbon dynamics, recalcitrant allochthonous carbon may decompose if disturbed[40]. Therefore, the trapping and burial of allochthonous carbon perhaps should be considered a part of blue carbon as it prevents the reintroduction of previously-stored carbon into the atmosphere. Furthermore, studies derived from historical SLR measurements are likely to underestimate the role of allochthonous carbon in future coastal ecosystems as both SLR rates and carbon exchange (Fig. 2) continue to increase. In the scenario presented here, allochthonous carbon contribution to marsh accretion quadruples as SLR increases from 2 mm yr⁻¹ to 7–15 mm yr⁻¹ (Fig. 2).

To test the importance of allochtonous carbon in the resilience of coastal carbon ecosystems, we conducted a third model experiment in which allochtonous carbon deposited on the marsh platform instantaneously mineralized (100% decomposition of allochtonous carbon, as opposed to 0% decomposition in previous experiments) so that it did not contribute to marsh elevation change or carbon cycling. Without allochtnous carbon accumulation, total CAR, autocthonous carbon, and marsh size is maximized at a lower rate of SLR [8 mm yr⁻¹] than for simulations with allochtonous carbon accumulation [11 mm yr⁻¹] (Fig. 3). Peak marsh width [~1700 m vs. ~1900 m] and peak total CAR [100 g m⁻² yr⁻¹ vs. 180 g m⁻² yr⁻¹] are lower compared to simulations with allochthonous carbon (Fig. 3). Although allochthonous carbon comprises less than half the carbon accumulation at very high SLR [10-15 mm yr⁻¹], marsh width and CAR are maintained by allocthonous carbon accumulation (Fig. 3a). Both marsh width and CAR decrease dramatically without allochthonous carbon (Fig. 3b). Furthermore, autochthonous marsh carbon is higher in simulations without allochthonous carbon and lower rates of SLR [1-8 mm yr⁻¹], driven by increased productivity from lower marsh elevations. However, at high rates of SLR [>8 mm yr⁻¹] autochthonous carbon cannot compensate for the lack of allochthonous carbon. At these high rates of SLR, elevations have decreased so as to lead to decreased plant production and less autochthonous accretion. This suggests that increased autochthonous carbon partially compensates for the reduction in allochthonous carbon, and emphasizes the complex nonlinear relationship between marshes and climate.

Furthermore, these model experiments give insight into our parameterization of organic matter decomposition and carbon lability. While in the first set of experiments all allochthonous carbon is refractory, these later experiments parametrize all allochthonous marsh carbon as labile with a very high decomposition rate (100% decomposes instantaneously). While the total amount of marsh carbon is sensitive to the amount of recalcitrant allochthonous carbon (indicated by differences between Fig. 3a and Fig. 3b), marsh extent and carbon storage peak at intermediate rates of SLR independent of the lability of allochthonous carbon. This highlights the underlying behavior of marshes and their ability to adapt to changing sea levels, independent of carbon lability parameterizations. However, the differences in the SLR tipping point and the total amount of carbon demonstrate the need to better understand carbon lability in coastal systems.

## Effect of SLR on landscape carbon budgets

Observations of carbon cycling in individual components of the coastal landscape demonstrate contrasting responses to SLR[16,17,66]. Furthermore, within marsh ecosystems field observations show both increases[61,63] and decreases[67,68] in marsh carbon storage with SLR. Overall, we find that the carbon stock summed across the width of the entire coastal landscape (bay-marsh-forest) increases with SLR up to an optimum rate of SLR [10 mm yr⁻¹], followed by a decline at faster rates, despite disparate responses within individual systems (Fig. 4). Our work corroborates previous ecosystem-specific findings by showing that forest carbon stocks decrease with SLR due to decreased tree biomass (Fig. 4). Bay-bottom carbon stocks increase with SLR, driven by increased accommodation space from increased water depth (Fig. 4). Little empirical work has been done to assess unvegetated bay-bottom contributions to coastal carbon cycling, despite their

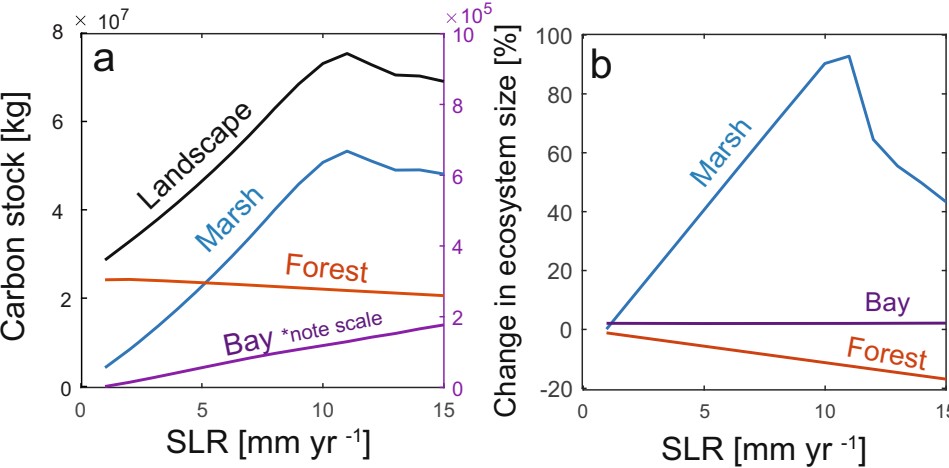

**Fig. 4 | Carbon stocks and ecosystem sizes depend on sea level rise (SLR) rates.** **a** Total landscape carbon stock, comprised of marsh, forest, and bay ecosystems vary with SLR. **b** Shifts in landscape carbon stocks depend on the size of each component of the landscape, where change in ecosystem size is relative to the initial size of the ecosystem. Carbon stocks were calculated at the end of the model experiments (100 years) and are the sum of both biomass and soil carbon.

importance for sediment exchange with other parts of the coastal system[32] and their potential for enhanced carbon storage[69,70]. Finally, the modeled marsh carbon stock is more complex, as it increases with SLR up to an optimum rate of SLR [10 mm yr⁻¹] and then declines. This nonlinear response may help explain the seemingly contradictory results observed in marshes, where both positive and negative relationships between SLR and CAR have been observed[63,67]. More specifically, our model results show mechanistically that the peak in total marsh carbon and total landscape carbon at an intermediate rate of SLR (Fig. 4a) is caused by the synchronous peaks in CAR, autochthonous CAR, and marsh size (Fig. 3a). This demonstrates that increased organic matter recycling, increased in-situ accumulation rates, and increased marsh size all contribute to increased landscape carbon storage.

The changing size of each component of the coastal landscape plays a primary role in determining how the total landscape-scale coastal carbon stock responds to SLR (Fig. 4). In our experiments, bay width remains relatively constant (bay size varies by 7 m), while forest width decreases by 100–1500 m [1–15 mm yr⁻¹ SLR], and vegetated marsh peaks at an intermediate rate of SLR (Fig. 4b). As SLR rates increase, there is a fundamental shift from forest-carbon dominated landscapes, where the majority of carbon [80% at 1 mm yr⁻¹ SLR] is stored as woody biomass, to marsh-carbon dominated landscapes, where more than 50% of the carbon is stored in the soils (Supplementary Fig. 6). Marsh carbon is more labile compared to forest carbon[71], meaning that the transition from a forest-dominated system to a marsh-dominated system represents a switch to more labile forms of carbon in the coastal landscape.

Interestingly, the total landscape carbon stock in the bay-marsh-forest system is higher at elevated rates of SLR compared to the landscape carbon stock at historical rates of SLR [>10 mm yr⁻¹ versus 1–2 mm yr⁻¹]. This observation suggests that the coastal landscape continues to store large stocks of carbon even as the size of marsh and forest ecosystems decline. Our model experiments indicate that the maintenance of high coastal carbon stocks is driven largely by enhanced connectivity between marshes and mudflats, so that marsh erosion leads to higher CAR at any remaining points on the marsh platform (Figs. 2, 3).

The importance of carbon connectivity highlighted on this generalized coastline can be extrapolated to other coastal marsh systems. For example, connectivity may be reduced in areas where marsh migration is hindered (urban development, steep upland slope). This restriction may result in decreased marsh extent and marsh carbon

storage. Similarly, increased erosion of the marsh edge (high winds, more exposed coastline) would increase the exchange of carbon across the bay-marsh interface. The increased connectivity from edge erosion would increase suspended sediment and allochthonous carbon adjacent to the marsh, resulting in higher CAR and enhanced marsh resilience to SLR. Likewise, increased tidal range increases the connectivity between the marsh and the bay, resulting in higher CAR and marsh extent (Supplementary Fig. 5). Our results underline the importance of connectivity for increased coastal resilience and carbon storage. While this model describes qualitative patterns in coastal landscape response to global change, it highlights the need for more robust couplings between interacting habitats in earth system models as we demonstrate these couplings fundamentally alter landscape carbon balances[27,72].

Ecosystem connectivity tends to increase the resilience and function of ecosystems responding to stressors in a variety of terrestrial and marine environments[23,24,28]. However, it remains unknown how climate change alters the impacts of connectivity on ecosystem function, particularly at large spatial scales such as coastal landforms. Our experiments uniquely reveal that climate change (i.e., SLR) increases connectivity between adjacent ecosystems (Fig. 2) in ways that enhance the function of the entire coastal landscape up to a point, after which the connectivity drives a decrease in ecosystem function. The increased connectivity and the maintenance of high coastal landscape carbon stocks come at a cost, even beyond the loss of marsh and forest ecosystems. Specifically, the transition from a coastal landscape dominated by forest carbon to one dominated by marsh carbon represents a switch to more labile carbon, and places more carbon in areas vulnerable to SLR and erosion, so that high carbon stocks become more precarious with increasing SLR and ecosystem connectivity. Thus, our work suggests that climate change enhances connectivity between coastal ecosystems, but with tradeoffs that become more negative under accelerated sea level rise.

## Methods
We developed CoLT (Coastal Landscape Transect model), a 2-D model of coastal landscape carbon cycling by coupling a geomorphic sediment transport model[31] with a point-based soil carbon accumulation model[29,30]. The model aims to capture carbon dynamics across a transect spanning a bay-marsh-forest coastal system (Fig. 1). Although flexibility in parameter choices could ultimately allow the simulation of a wide-variety of coastal settings, our model is most explicitly designed to consider the evolution of a gently sloping coastal plain (0.001) with

moderate sediment inputs and regular tides, where changes in marsh width are driven by the balance between vertical accretion and sea level rise, and the balance between marsh edge erosion and migration into adjacent upland forests. The geomorphic components of the model are based on the transect model by Kirwan and others[31] that simulates the transport of sediment and migration of coastal ecosystems through space and time. The carbon accumulation components of the model are based on the soil cohort model by Kirwan and Mudd[29] and Rietl et al.[30] that simulates carbon production and decomposition at a given point on the marsh platform, and with depth in the soil profile. Each cell is 1 m wide and the timestep used throughout the model experiments is one year.

## Geomorphic processes

Following previous approaches[31,36], the lateral position of the marsh-bay boundary is treated as the difference between seaward marsh progradation and landward marsh erosion. Wind speed, fetch, and water depth affect the wave properties[73], which in turn affects the erosion rate. The wave power density, $W = \frac{1}{16}\gamma c_g H_s^2$, is related to the edge erosion ($B_e$) as:

$$B_e = k_e W \qquad (1)$$

where $k_e$ is a fitting coefficient that is related to the vegetation and sediment characteristics (i.e., erodibility), $\gamma$ is specific weight of water, $c_g$ is the group wave velocity, and $H_s$ is the significant wave height. Marsh progradation ($B_p$) is a function of suspended sediment at the marsh edge ($C_r$), sediment bulk density ($\rho$), settling velocity ($w_{sf}$), and an empirical coefficient ($k_a$):

$$B_p = k_a w_{sf} \rho^{-1} C_r \qquad (2)$$

In our model simulations, $k_a$, $w_{sf}$, and $\rho$ are set to constants (Supplementary Table 1), and $C_r$ is determined by the bed shear stress. Here, bed shear stress, $\tau_w$, is composed only of energy from waves. The overall change of the bay-marsh boundary is the balance between $B_e$ and $B_p$. The bay-marsh boundary is not stable[74] and therefore this balance is almost never zero. The migration of the bay-marsh boundary changes the fetch and therefore changes the wave conditions throughout the simulation.

The bay depth evolves dynamically throughout the model simulation depending on sediment availability (internal from the marsh and external from the outside of the bay) and the size of the bay (Mariotti and Carr, 2014). In our simulations, we use an external suspended sediment supply of 50 mg L$^{-1}$ which represents the delivery of sediment from the ocean side of the domain, such as inlet exchange, redistribution of continental shelf sediments, or river plumes. The internal sediment supply is a function of both the amount of material eroded from the marsh edge (balance between $B_e$ and $B_p$) and the amount of sediment resuspended from the bay bottom (function of the excess shear stress, $\tau = \max\left(\frac{\tau_w - \tau_{cr}}{\tau_{cr}}, 0\right) * \lambda$, where $\tau_w$ is the wave-generated shear stress, $\tau_{cr}$ is the critical shear stress, and $\lambda$ is a coefficient representing bay sediment erodibility). For sediment delivery to the marsh, we calculate the concentration of sediment near the marsh edge based on edge erosion and bed resuspension. For more details on the bay bottom evolution, see Mariotti and Carr[36]. While the model does simulate changes in wave height related to changes in the bay depth and fetch, those changes are relatively small and result in minor changes to lateral retreat rate in the simulations presented here.

In order to maintain their vertical position in the tidal frame, salt marshes accrete both mineral ($a_m$) and organic matter ($a_o$) as $\frac{dz}{dt} = (a_m + a_o)/\rho$. The mass of mineral sediment deposition depends on the suspended sediment concentration and the settling velocity of the sediment particles over the duration of inundation. The deposition of these particles is not constant over the marsh platform. Instead it

decreases exponentially with distance from the marsh edge according to:

$$C_x = C_r e^{-\Lambda x} \qquad (3)$$

where $C_x$ is the suspended sediment concentration at distance $X$, $C_r$ is the suspended sediment concentration at the marsh edge, and $\Lambda$ is a decay coefficient[31].

The organic component depends on the belowground biomass of the plants, $B$, which is a quadratic function that relates the marsh depth during inundation, $d$, to plant productivity[29]:

$$B = \frac{4B_{max}(d - d_{max})(d - d_{min})}{(-d_{min} - d_{max})(d_{max} - 3d_{min})} \qquad (4)$$

Where $B_{max}$ is the peak biomass, $d_{max}$ is the maximum depth that plants can grow, $d_{min}$ is the minimum depth at which the plants can grow. This quadratic relationship is most representative of *Spartina alterniflora*[50] and has been widely used in ecogeomorphic models of marsh evolution[31,36,38,75–78].

As sea level rises, marsh systems migrate into the upland forests. The location of the marsh-forest boundary is dictated by the slope of the uplands[31,79] and the rate of SLR. We use the simple model described by Kirwan et al.[31], which assumes the passive and continual upland migration of the marsh-forest boundary ($B_l$), $B_l = R/m$, where $R$ is the SLR rate and $m$ is the upland slope, which is kept constant (0.001, coastal plain[31]) in our simulations.

## Carbon processes

The bay bottom sediment is set to have an initial organic carbon content of 5%[80]. This OC is considered allochthonous, as there is no primary production on the bay bottom in the model and therefore the carbon must have been produced elsewhere. Given that it is allochthonous and centuries to millennia old, we assume that this carbon does not decompose. Any sediment imported into the bay (i.e., external sediment supply) has an OC content equal to that of the bay bottom, representing the organic carbon attached to mud particles[80]. In addition, organic carbon eroded from the marsh edge deposits on the bay bottom, adding carbon to the sediments. The carbon is distributed equally across the bay bottom, and once it enters the bay is considered allochthonous and therefore does not decompose[41].

The formulation for marsh carbon dynamics follows a soil-cohort approach, in which organic matter accumulates in layers of soil as the balance between productivity and decomposition[29,30]. Both aboveground and belowground biomass is modeled as a quadratic function of marsh elevation[50]. We set belowground and aboveground biomass ($B$) to be equal, with biomass maximized at an intermediate elevation. The aboveground biomass is included in estimates of carbon stocks but does not contribute to organic vertical accretion. Allochthonous organic material is deposited on the marsh surface, while autochthonous carbon is distributed with depth in an approach similar to Rietl et al.[30]. Autochthonous organic matter undergoes depth-dependent decomposition using a soil cohort approach according to:

$$\text{decomp} = \text{OM}_{auto} * m_k * e^{-\text{depth}/m_u} \qquad (5)$$

where $\text{OM}_{auto}$ is the amount of autochthonous organic matter in a given layer of sediment, depth is the depth of the given layer of sediment, $m_k$ is the coefficient of decomposition and $m_u$ is the depth at which decomposition goes to zero. Following previous approaches[29,30], this gives rise to a relationship in which most decomposition happens at the marsh surface and decomposition decreases with depth. In addition, there is an allochthonous component of organic matter that deposits on the marsh surface as a function of distance from the marsh edge, as it is delivered from marsh flooding

in the same way that the mineral sediment is delivered to the marsh. As allochthonous material is often millennia old and thought to be composed entirely of recalcitrant material, it does not decompose in the model. Within each annual soil cohort, the model calculates the bulk density of the sediment, which changes through time as organic matter is produced and decomposed. Following decomposition, the marsh elevation is updated to reflect the decrease in elevation.

As the marsh edge erodes, the mass of mineral and organic material from each eroded soil cohort is summed to determine the amount of mineral and organic material that is transported to the bay and therefore available for redeposition on the marsh. The organic content of the bay sediment evolves dynamically balancing inputs and exports of organic matter, including: the input of organic content from the eroded marsh edge sediment, the input of organic content from the external sediment supply, and the export of organic material delivered to the marsh. Thus, the total amount of organic material deposited on the marsh depends on the organic content of the bay.

To be able to compare carbon stocks between systems as marsh transgresses into forest, we developed carbon models (both aboveground and belowground) for the coastal forest system. Aboveground biomass in the forest is modeled as a logarithmic function of elevation, where production increases with higher elevations up to a maximum carrying capacity. This is supported by field measurements[17] and remote-sensing observations (Supplementary Fig. 7).

$$C(z) = \frac{B_{\max\,forest}}{(1 + a \exp(-bz))} \quad (6)$$

Or

$$\frac{dC}{dz} = \frac{b\left(B_{\max\,forest} - C\right)}{B_{\max\,forest}} C \quad (7)$$

Where $a = (B_{\max\,forest} - C_0)/C_0$, representing the starting value at the forest edge, $C$ is the biomass of the trees (including roots), $C_{0,agb}$ is the amount of carbon in the transition zone from trees, and $b$ is the growth rate. This parameterization represents the gradual death of trees as they experience increased stress from flooding and saltwater intrusion.

We model the belowground carbon stock using two processes. First, there is a very low rate of carbon deposition, representing the carbon accumulation in the soils from the forest itself. We assume that the carbon has already undergone decomposition when it has deposited. Based on field data[17], we also note that there is a thicker organic layer on the forest floor at lower elevations. This is likely due to more saturated conditions that allow organic matter to build up. We therefore impose an elevation-dependent gradient in carbon deposition in the forest that sums with the constant background[17]. Therefore, the belowground forest carbon is formulated as:

$$C(z) = -C_{wet} \exp(-b_{soil}z) + C_{0,soil} \quad (8)$$

Where $C$ is the carbon deposited in a given year, $C_{0,soil}$ is the background carbon accumulation in the soils across the entire forest, $z$ is elevation, $b_{soil}$ is a decay constant, and $C_{wet}$ as the carbon layer from wetted soils.

## Model experiment setup

All model runs began with a 5 km wide bay of equal depth, a 1 km wide marsh of equal elevation, and an 8 km wide coastal forest. The forest width was designed to accommodate the maximum landward migration of the marsh for the highest SLR scenario. The initial marsh platform was developed with a spinup period of 550 years with a SLR of 1 mm yr$^{-1}$, in which layers of organic matter were deposited. Initial marsh width was imposed to 1 km, and the spinup resulted in a marsh

with one elevation (i.e., no topography). The low rate of SLR used represents the historical SLR rate. All mud in the bay was modeled to have an initial carbon content of 5%[80]. Following the spinup period, we modeled the evolution of the coastal transect under a range of SLR scenarios [1–15 mm yr$^{-1}$] with a moderate sediment supply (50 mg L$^{-1}$) for 100 years. All scenarios began with the underlying stratigraphy and elevation profile from the spinup [1 mm yr$^{-1}$] and SLR was instantaneously changed to the 15 different scenarios (1–15 mm yr$^{-1}$). We use the same constants for all presented model runs (Supplementary Table 1), including tidal range, suspended sediment supply, upland slope, and wind speed, to demonstrate that differences between model simulations represent the response to changes in SLR.

## Sensitivity analysis

To test model sensitivity to key environmental parameters driving marsh evolution, we ran the model for a wide range of input suspended sediment concentrations and tidal ranges. Sediment supply is a key driver of the evolution of natural marshes[81,82], and the suspended sediment concentration (SSC) of the bay is a key parameter affecting marsh sustainability in numerical models[77,83]. Under low to moderate rates of SLR [1–6 mm yr$^{-1}$], SSC input did not substantially change any of the key model results we explored (Supplementary Fig. 3). At higher rates of SLR [>6 mm yr$^{-1}$] and intermediate SSC [20–90 mg L$^{-1}$], we also observed no substantial changes in the metrics used in our study. However, at both high rates of SLR and extremely small or large SSC, the model results are substantially impacted. At low SSC [10 mg L$^{-1}$], marsh width, CAR, and marsh C are reduced, while at high rates of SLR and SSC = 100 mg L$^{-1}$, marsh width is increased. Forest C is not affected by changes in SSC. These results indicate that the model is not overly sensitive to SSC, and that consistent results are obtained over a wide range of reasonable SSC [i.e., 20–90 mg L$^{-1}$]. Within this range of SSC [20–90 mg L$^{-1}$], we further observe synchronous peaks in marsh width, CAR, and autochthonous CAR at intermediate rates of SLR (Supplementary Fig. 4). Together, this sensitivity analysis highlights that landscape carbon is driven mechanistically by synchronous peaks in CAR and marsh width, independent of the external sediment supply.

Another dominant factor in marsh evolution is tidal range[77,83]. To test the sensitivity to tidal range, we ran the model using two tidal ranges (1 and 3 m), in addition to the 1.4 m tidal range considered in the simulations presented in the main text. These results demonstrate that allochthonous carbon deposition increases with tidal range (Supplementary Fig. 5). Like the original simulations, marsh width and CAR tend to increase with SLR towards an optimum SLR rate regardless of tidal range. However, with a large tidal range, only the rising limb of marsh extent and CAR are observed (Supplementary Fig. 5). We attribute this pattern to the lack of marsh drowning in the experiment, driven by spinup conditions that created a marsh that was initially higher in elevation, and the well-known link between tidal range and marsh sustainability[77,83].

## Data availability

All model simulations generated during and/or analyzed during the current study can be recreated using the code available in the CSDMS model repository (https://doi.org/10.5281/zenodo.7625873). Specific model simulations can be requested from the corresponding author.

## Code availability

Model code is available on the CSDMS model repository: https://csdms.colorado.edu/wiki/Model:Coastal_Landscape_Transect_Model_(CoLT) (https://doi.org/10.5281/zenodo.7625873), as well as from the corresponding author on request.

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

## Acknowledgements

The authors thank A. Cole for support on this project and I.V. Cole for feedback on figures. The work was supported by the NSF CAREER Award (1654374, M.L.K.). Additional support was provided by the National Science Foundation (LTER 1237733, M.L.K.), the Department of Energy (DE-SC0014413, DE-SC0019110, and DE-SC0021112, M.L.K.), and the USGS Climate Research and Development and Ecosystems Programs (D.C.W.). Any use of trade, firm, or product names is for descriptive purposes only and does not imply endorsement by the U.S. Government.

## Author contributions

K.V. coded the model, did the analysis, and wrote the manuscript. E.R.H. and D.C.W. coded part of the model. Y.C. and A.J.S. aided in data-model integration. M.L.K. secured funding. K.V., E.R.H., D.C.W., and M.L.K. all contributed to ideas and project conceptualization. All authors provided edits and comments on the manuscript.

## Competing interests

The authors declare no competing interests.
