## [Peer Review File · Nature Communications]

Climate-driven tradeoffs between landscape connectivity and the maintenance of the coastal carbon sinkReviewer #1 (Remarks to the Author):

This modeling study investigates the role of ecosystem connectivity in mediating coastal ecosystem function and the coastal carbon sink in response to sea-level rise (SLR), where ecosystem connectivity is defined as the transfer of material and carbon across the landscape. Specifically, a coupled spatially-explicit geomorphic model with a point-based carbon accumulation model of a bay-marsh-forest transect is developed and used to quantify carbon transfer and accumulation across a range of SLR scenarios. Simulations show that, despite divergent responses of individual ecosystem carbon stocks to SLR, the integrated coastal zone carbon sink peaks at intermediate SLR scenarios. However, this peak carbon may be more vulnerable to erosion and decomposition due to the transfer from forest biomass to marsh soil carbon.

This study touches on an important and growing topic in coastal zone sciences: the integrated coastal system and links between subtidal, intertidal, and upland systems. I am excited to see this study's focus on ecosystem connectivity, as studies have addressed similar questions in individual systems or between two systems without considering the whole coastal landscape. Results highlight a goldilocks behavior where moderate rates of SLR may actually enhance coastal carbon stocks, but stocks will decline following a landscape-mediated tipping point. I enjoyed reading this study but have a few comments on the study approach, text, and depiction of results as articulated below.

Sincerely,

General Comments:

- 1) I appreciate the model developments and the complexity of the integrated landscape with a coupled geomorphic and carbon model – a significant step in itself. I also recognize the limitations of coupled model systems. However, this study overlooks an important form of connectivity in the coastal zone: hydrologic connectivity. Groundwater impacts lateral carbon transport between the marsh and the bay, as well as the vegetation dynamics in the marsh and forest. I am not advocating for more simulations here nor further model developments (although I think that could be a natural future step), but it would be beneficial to add a sentence to the introduction and/or discussion to acknowledge this ecosystem component and the potential impact of groundwater on ecosystem connectivity, carbon stocks and the coastal landscape trajectory.**
- 2) Similar to the comment above, it would be beneficial to add text addressing caveats and model limitations. Inherently, generalized models incorporate numerous simplifications. I see the immense value in these generalized models, but I think it is important to discuss simplifications and potential impacts on model results. I suggest adding a few sentences discussing the model limitations and simplifications and potential impacts on model projections. For example, with sea-level rise, air and water temperatures will change with implications for productivity and decomposition. How may this additional factor impact the results, or would it?
 - a. I've also pointed out areas where additional justification would be beneficial in the 'specific comments' below.****
- 3) The color scheme in Figure 1 b,c is confusing. At first, I thought the blue represented water depth given the blue color and the closeness to the bay color in Figure 1a. I suggest changing the color gradient of allochthonous carbon to brown-to-red or deep green-to-red (or something along those lines). Also, in Figure 1b, it looks like the bay both expands and contracts based on the dark grey line (bay) and light gray section (marsh). Add text in caption to clarify or correct.**
- 4) The text is well written, but please be mindful of long, cascading sentences. I've identified sentences that are particularly long below.**

Specific Comments:

Line 10: If word count allows, I suggest adding a clause that defines ecosystem connectivity.

Line 28: Remove the first 'carbon' as it is repeated.

Line 41: I suggest removing 'the carbon exchange is' so the sentence reads "Exchange... is substantial but poorly quantified...".

Line 64: The second sentence in this paragraph (starting with 'Previous') is long and cascading, making it tough to follow. I suggest separating this into multiple sentences.

Line 71: Similar to above comment, I suggest separating this sentence into multiple or rearranging for clarity and succinctness.

Line 84: What about seagrass beds? I suggest adding a clarifying clause or sentence here about the type of bay or depth to justify no bottom vegetation.

Line 85: Groundwater discharge can also be a source of carbon to the bay. Again, I understand this is not part of an already complex model, but this would be a good location to address groundwater, its role, and that it is not explicitly part of this model study.

Line 90: Please add a reference to the end of this sentence.

Model approach and basic behavior section: How is inundation depth sea-level integrated into the model? It would be beneficial to add a sentence here to explicitly state where and how the sea level at a given year drives the simulations.

Line 130: The first two sentences of this paragraph are repetitive and can be combined.

Line 140: Please split into two sentences.

Line 143: It may be beneficial to ground the SLR rates to specific coastlines. For example, add a clause along the lines of 'as projected along Chesapeake Bay coastlines'.

Line 160: Consider adding a clause defining recycled carbon.

Figure 2: What is the peak in C eroded at ~14 mm/yr SLR due to?

Line 186: Remove 'however'.

Line 211: Did you consider altering the 50 mg/L as well as the complete mineralization of allochthonous carbon? If I understand correctly, this could address the relative importance of edge erosion vs. bay inputs.

Line 232: There first three sentences are repetitive with introduction. Given word limitations, this can be reduced to a sentence or two.

Line 238: Is there any time consideration for tree mortality in the model, given that tree mortality is often slow?

Line 245: So, would different ecosystems with different initial marsh widths peak at different times/SLR rates? I suggest adding some discussion text addressing the results highlighted in these last two sentences of the paragraph.

Line 259: This sentence contradicts Figure 4A. Please clarify.

Line 327: Does the sediment bulk density vary with depth (i.e., is sediment compaction considered in the model)?

Line 326-330 & Line ~370: Of the material that is eroded and redeposited, what percent is organic matter? How is it determined?

Reviewer #2 (Remarks to the Author):

The ms is not ready for reviewing.

The modeling relies on many parameters, some are empirical, some are derived from observations. Each of them has error bars. The model predictions thus also have error bars. We need to know these error bars in the values of the parameters and in the model output before we can review this manuscript in order to judge the reliability of the model predictions.

Reviewer #1 (Remarks to the Author):

This modeling study investigates the role of ecosystem connectivity in mediating coastal ecosystem function and the coastal carbon sink in response to sea-level rise (SLR), where ecosystem connectivity is defined as the transfer of material and carbon across the landscape. Specifically, a coupled spatially-explicit geomorphic model with a point-based carbon accumulation model of a bay-marsh-forest transect is developed and used to quantify carbon transfer and accumulation across a range of SLR scenarios. Simulations show that, despite divergent responses of individual ecosystem carbon stocks to SLR, the integrated coastal zone carbon sink peaks at intermediate SLR scenarios. However, this peak carbon may be more vulnerable to erosion and decomposition due to the transfer from forest biomass to marsh soil carbon.

This study touches on an important and growing topic in coastal zone sciences: the integrated coastal system and links between subtidal, intertidal, and upland systems. I am excited to see this study's focus on ecosystem connectivity, as studies have addressed similar questions in individual systems or between two systems without considering the whole coastal landscape. Results highlight a goldilocks behavior where moderate rates of SLR may actually enhance coastal carbon stocks, but stocks will decline following a landscape-mediated tipping point. I enjoyed reading this study but have a few comments on the study approach, text, and depiction of results as articulated below.

Sincerely,

Thank you very much for the thoughtful review and the insightful comments to help improve this study. We carefully addressed each of your comments and updated the manuscript accordingly. Please see our detailed responses below or refer to our revised manuscript with tracking for the exact edits.

General Comments:

1) I appreciate the model developments and the complexity of the integrated landscape with a coupled geomorphic and carbon model – a significant step in itself. I also recognize the limitations of coupled model systems. However, this study overlooks an important form of connectivity in the coastal zone: hydrologic connectivity. Groundwater impacts lateral carbon transport between the marsh and the bay, as well as the vegetation dynamics in the marsh and forest. I am not advocating for more simulations here nor further model developments (although I think that could be a natural future step), but it would be beneficial to add a sentence to the introduction and/or discussion to acknowledge this ecosystem component and the potential impact of groundwater on ecosystem connectivity, carbon stocks and the coastal landscape trajectory.

Thank you for the kind words. We agree that we have neglected groundwater, hydrologic connectivity, and its impact on carbon transport between ecosystems. In response, we have noted this omission and it certainly could be a next step for the development of this model.

We explicitly pointed out this limitation by adding the following in the manuscript:

“Furthermore, we limit carbon processes to particulate carbon that is directly associated with sediment transport processes and biomass production. The outwelling of carbon, in both dissolved and particulate

form, from groundwater, is an important part of the carbon budget in the coastal zone⁴³ and for plant productivity⁴⁴, but is not resolved here.”

2) Similar to the comment above, it would be beneficial to add text addressing caveats and model limitations. Inherently, generalized models incorporate numerous simplifications. I see the immense value in these generalized models, but I think it is important to discuss simplifications and potential impacts on model results. I suggest adding a few sentences discussing the model limitations and simplifications and potential impacts on model projections. For example, with sea-level rise, air and water temperatures will change with implications for productivity and decomposition. How may this additional factor impact the results, or would it?

This is a good point. Model limitations are peppered throughout the article now, including those about seagrasses (see response to your later comment), other climate change effects, and groundwater. Some of the effects are nonlinear, so the effects on the results may be specific to parameters used. We described the simplifications, and addressed their limitations in the manuscript as below:

“Although both autochthonous carbon production and decomposition will be altered with a changing climate^{41,42}, the nonlinear interactions between temperature, CO₂, and nutrients are complex and beyond the scope of this model. Therefore, we do not capture all climate-carbon feedbacks but focus solely on the climate effect of SLR. Furthermore, we limit carbon processes to particulate carbon that is directly associated with sediment transport processes and biomass production. The outwelling of carbon, in both dissolved and particulate form, from groundwater, is an important part of the carbon budget in the coastal zone⁴³ and for plant productivity⁴⁴, but is not resolved here.”

a. I’ve also pointed out areas where additional justification would be beneficial in the ‘specific comments’ below.

Thank you for these specific comments. We made sure to address each of them specifically (see responses below). In particular, we added justification for assumptions or simplifications in the model, including the lack of groundwater, nonlinear effects with climate change, and the absence of seagrass. In response to your comments, we also added discussion about tree mortality.

3) The color scheme in Figure 1 b,c is confusing. At first, I thought the blue represented water depth given the blue color and the closeness to the bay color in Figure 1a. I suggest changing the color gradient of allochthonous carbon to brown-to-red or deep green-to-red (or something along those lines). Also, in Figure 1b, it looks like the bay both expands and contracts based on the dark grey line (bay) and light gray section (marsh). Add text in caption to clarify or correct.

We changed the color scheme in 1b and c to brown-to-red as suggested to avoid any water confusion. We also adjusted the shapes for the bay to make 1b clearer.

4) The text is well written, but please be mindful of long, cascading sentences. I’ve identified sentences that are particularly long below.

Thanks for pointing them out. We reworked the sentences that were particularly problematic as suggested.

Specific Comments:

Line 10: If word count allows, I suggest adding a clause that defines ecosystem connectivity.

Because of word limits, we prioritized edits that made the model clearer and did not add a definition of connectivity in the summary. We think it would be a good addition, but we just don't have the space! The last paragraph of the introduction in the main gives a more robust definition.

Line 28: Remove the first 'carbon' as it is repeated.

Changed

Line 41: I suggest removing 'the carbon exchange is' so the sentence reads "Exchange... is substantial but poorly quantified...".

Changed

Line 64: The second sentence in this paragraph (starting with 'Previous') is long and cascading, making it tough to follow. I suggest separating this into multiple sentences.

Rewritten to be:

"Previous work typically models the geomorphology or carbon storage processes of an individual system^{29,30}. In cases where multiple systems are considered, either the geomorphology^{31,32} or carbon storage³³ is modeled, but not both."

Line 71: Similar to above comment, I suggest separating this sentence into multiple or rearranging for clarity and succinctness.

The sentence was split into three sentences for improved clarity, and they now read:

"We started with a geomorphic transect model that considers the topographic evolution of bay bottoms, marshes, and forests³¹. The bay width and depth evolve as a function of wind speed, fetch, and water depth³⁶. Marshes accrete as a function of mineral sediment deposition and biomass production²⁹. Forests migrate passively as a function of upland slope and SLR³¹."

Line 84: What about seagrass beds? I suggest adding a clarifying clause or sentence here about the type of bay or depth to justify no bottom vegetation.

We have added the following sentences:

"While some shallow bays are vegetated with seagrass³⁸, they induce further geomorphic feedbacks. For example, seagrass colonization depends on the elevation of the bay bottom; however, this model represents the bay bottom with a single elevation."

Line 85: Groundwater discharge can also be a source of carbon to the bay. Again, I understand this is not part of an already complex model, but this would be a good location to address groundwater, its role, and that it is not explicitly part of this model study.

Very good point. To address your concern, we added the following:

“Furthermore, we limit carbon processes to particulate carbon that is directly associated with sediment transport processes and biomass production. The outwelling of carbon, in both dissolved and particulate form, from groundwater, is an important part of the carbon budget in the coastal zone⁴³ and for plant productivity⁴⁴, but is not resolved here.”

Line 90: Please add a reference to the end of this sentence.

The van de Broek et al. (2018) citation within the same sentence demonstrates this point well, and so we included this citation again to support this point.

Model approach and basic behavior section: How is inundation depth sea-level integrated into the model? It would be beneficial to add a sentence here to explicitly state where and how the sea level at a given year drives the simulations.

Added the following:

“Sea level is calculated yearly and used to determine marsh inundation, and therefore mineral and organic sediment deposition, as well as plant biomass and therefore carbon production, each year.”

Line 130: The first two sentences of this paragraph are repetitive and can be combined.

Rewritten to be as follows:

“We also observe that the mass of allochthonous carbon is highest near the marsh edge in all scenarios and that autochthonous carbon dominates the marsh interior (Fig. 1b, 1c).”

Line 140: Please split into two sentences.

Changed

Line 143: It may be beneficial to ground the SLR rates to specific coastlines. For example, add a clause along the lines of ‘as projected along Chesapeake Bay coastlines’.

As this model is exploratory, we wanted to be clear that we are not representing a specific coastline and instead representing an idealized one. Therefore, we ran experiments that represented the entire possibility space of SLR rates, from historical to those over the next century (consistent with IPCC). We clarified this in the text to explain why we ran these simulations as follows:

“To explore the impacts of connectivity on coastal carbon, we conducted a second set of experiments under a larger range of SLR rates that represent the range of potential SLR rates in the next century [1-15 mm yr⁻¹].”

Line 160: Consider adding a clause defining recycled carbon.

Added

“At this point, the amount of recycled carbon (i.e. carbon eroded from the marsh and then redeposited on the marsh) becomes increasingly important for deposition on the marsh surface (Fig. 2b), but the amount of material eroded from the marsh edge cannot sustain long-term marsh growth.”

Figure 2: What is the peak in C eroded at ~14 mm/yr SLR due to?

We don't fully understand the peak in C eroded at ~ 14 mm/yr, but believe it is a model artifact related to large-scale drowning of the marsh interior. The model presented here is a transect model, so by nature it cannot properly resolve creeks and ponds, which drive heterogeneity across a marsh platform in natural marshes. Because of this, some of the behavior at high rates of SLR (13-15 mm/yr) is unstable when ponds would become numerous and/or drain by expanding tidal channels during the drowning process. In response to Reviewer #2, we added a sensitivity analysis of model behavior under a variety of suspended sediment concentrations. The resulting envelope of model results illustrates that eroded C generally increases with SLR. Interestingly, the peak in eroded C at 14 mm/yr is not evident in the envelope of model results, indicating that it does not persist at relatively low and relatively high suspended sediment concentrations. Therefore, we believe that it is more proper to emphasize the general model trends in the manuscript discussion.

In the text, we added the following:

“We do not explicitly model ponds or channel development, as the model is a transect.”

Line 186: Remove ‘however’.

Changed

Line 211: Did you consider altering the 50 mg/L as well as the complete mineralization of allochthonous carbon? If I understand correctly, this could address the relative importance of edge erosion vs. bay inputs.

The original manuscript considered an experiment with and without allochthonous carbon, where the “no allochthonous carbon” case represents the complete mineralization of allochthonous carbon. We have added a phrase to the section of the manuscript that discusses those experiments to clarify that the experiment explicitly quantifies the effects of complete mineralization of allochthonous carbon. This now reads:

“To test the importance of allochthonous carbon in the resilience of coastal carbon ecosystems, we conducted a third model experiment in which allochthonous carbon deposited on the marsh platform instantaneously mineralized (100% decomposition of allochthonous carbon, as opposed to 0% decomposition in previous experiments) so that it did not contribute to marsh elevation change or carbon cycling.”

In addition, we have added a full sensitivity analysis on the effects of external sediment supply in response to Reviewer #2 (see response to reviewer #2 and the revised Figure 2 for details). Briefly, that sensitivity analysis found that the relationship between allochthonous carbon and SLR (Figure 2b, Supplementary Figure 2) holds for a wide variety of suspended sediment concentrations (SSC = 20 – 90 mg/L). We have added two sentences to the manuscript stating that the general relationship holds for a range of SSC.

To the model approach section:

“We performed a sensitivity analysis to demonstrate that the conceptual insights hold across a wide parameter space (Supplementary Figs. 2-4).”

To the connectivity section:

“This general relationship holds for a range of external sediment supplies (Supplementary Figs. 2-4).”

Finally, we conducted a sensitivity experiment where the complete mineralization of allochthonous carbon and the SSC are varied at the same time (Figure R4). This is added below in response to reviewer 2 as well, but since you specifically asked, we have included the same text here. This analysis provides interesting insights, but go beyond the scope of this paper, so we did not add this component to the main text.

We also tested model sensitivity to external sediment supply in the absence of allochthonous carbon deposition. When all allochthonous carbon is completely mineralized, the results (marsh extent, carbon) are more sensitive to external sediment supply (Fig. R4). The presence of allochthonous carbon is a buffer for marshes, and allows the basic underlying patterns of CAR to persist under a wide range of conditions. Without allochthonous carbon, the CAR patterns are strongly affected by low sediment supply (10-30 mg L⁻¹) across the entire range of SLR tested, indicating that without external sediment supply and allochthonous C delivery, the patterns of marsh carbon accumulation are altered. Namely, CAR and marsh C are higher at low SSC and SLR, relative to the model results presented in the paper, driven by increases in autochthonous production. However, marsh width and total forest carbon are not substantially affected by changes in external sediment supply (Fig. R4a and R4d). Marsh width is only affected at high rates of SLR and extremely large or small sediment supplies [10 or 100 mg L⁻¹], while patterns in forest carbon are insensitive to these variables. These analyses confirm a consistent peak in marsh width and CAR at intermediate rates of SLR, and demonstrate the importance of allochthonous C.

Figure R4. Relative change in marsh width (a), carbon accumulation rate (b), total marsh carbon (c), and total forest carbon (d) for a range of suspended sediment concentrations and SLR rates in a scenario with no allochthonous carbon deposition. Relative change was calculated as the difference between model runs presented in the main text (Figs. 3a-4, SSC=50 mg L⁻¹, SLR = 1-15 mm yr⁻¹) and model runs presented here (SSC=10-100 mg L⁻¹, SLR = 1-15 mm yr⁻¹). The vertical box in each figure indicates the model runs in the main text.

Line 232: There first three sentences are repetitive with introduction. Given word limitations, this can be reduced to a sentence or two.

Reduced to:

Observations of carbon cycling in individual components of the coastal landscape demonstrate contrasting responses to SLR^{16,17,57}. Furthermore, within marsh ecosystems field observations show both increases^{52,54} and decreases^{58,59} in marsh carbon storage with SLR.

Line 238: Is there any time consideration for tree mortality in the model, given that tree mortality is often slow?

Tree death is driven by both press and pulse disturbances (Kirwan and Gedan 2019), which can lead to stochastic retreat rates on the yearly time scale. However, on the centennial time-scale (on which this model is run) lateral forest migration rates are closely linked to relative sea level rise rates and show clear

acceleration with accelerating rates of SLR (Kirwan and Gedan 2019; Schieder and Kirwan 2019). Therefore, while individual tree death may be slow, lateral migration rates depend on the rate of SLR.

The gradual death of trees, and therefore reduction in forest carbon (i.e. tree biomass carbon) is represented in the parameterization of tree biomass. We represented tree biomass as a logistic growth curve (based on field and remote sensing work), where tree biomass is low near the forest edge and increases with distance from the marsh. This represents the gradual death of the trees as they are more impacted by salt water intrusion and flooding. By the time the forest transitions to marsh, the majority of tree biomass has decomposed. Dead trees which may persist at the marsh-forest boundary make up a small portion of the total carbon (<9%, K.Krauss et al. 2018). A potential expansion of this model (beyond the scope of this study) would be to try to represent any forest carbon preservation that occurs during marsh migration, but we expect this value to be small.

The following was added to the methods section describing the forest carbon:

“This parameterization represents the gradual death of trees as they experience increased stress from flooding and salt water intrusion.”

Line 245: So, would different ecosystems with different initial marsh widths peak at different times/SLR rates? I suggest adding some discussion text addressing the results highlighted in these last two sentences of the paragraph.

This is an interesting question. We ran additional model simulations with a smaller marsh (500 m wide). We see with an initial marsh width of 500m, the peaks in CAR and marsh width do shift (peaks at a higher rate of SLR), but we continue to observe synchronous peaks in CAR (total and autochthonous) and marsh width (Fig. R1). Therefore, although we observe a shift in the peaks, the results still indicate that peaks in landscape carbon storage is mechanistically driven by synchronous peaks in CAR and marsh width. Furthermore, we also observe the same pattern in the parameters – CAR and marsh width increase to a certain point and then decrease. For space reasons, and given that the main point of that paragraph is the synchronous peaks, rather than the rate of SLR at which the peaks occur, we did not add text to the manuscript, but provide the evidence here to support our findings.

Figure R1: Carbon accumulation rate and marsh width after the end of a 100-year model simulation with an initial marsh width of 1000 m (A), as in the model results presented in the main text, and 500 m (B), both with SSC=50 mg/L. Results indicate synchronous peaks in CAR and marsh width.

Line 259: This sentence contradicts Figure 4A. Please clarify.

Rephrased for clarity:

“Interestingly, the total landscape carbon stock in the bay-marsh-forest system is higher at elevated rates of SLR compared to the landscape carbon stock at historical rates of SLR [$>10 \text{ mm yr}^{-1}$ versus $1\text{-}2 \text{ mm yr}^{-1}$].”

Line 327: Does the sediment bulk density vary with depth (i.e., is sediment compaction considered in the model)?

Added the following:

“Within each annual soil cohort, the model calculates the bulk density of the sediment, which changes through time as organic matter is produced and decomposed.”

Line 326-330 & Line ~370: Of the material that is eroded and redeposited, what percent is organic matter? How is it determined?

The amount of organic matter is calculated dynamically, depending on the amount of mineral and organic material at the marsh edge at the given time. The amount of each proportion is tracked for each soil cohort and then the amount of mineral and organic material liberated during erosion reflects the sum of these components the marsh edge.

The organic content of the bay is determined as the combination of the organic content of the material eroded from the marsh edge (calculated dynamically depending on the organic content of the marsh edge

cell), the organic content of the material deposited on the marsh edge (dynamically calculated depending on the organic content of the bay bottom (i.e. resuspension of bay bottom sediments) from the previous time step), and the organic content of the external sediment supply (constant). Therefore, if the marsh edge is more organic-rich, it increases the organic content of the bay bottom and the resulting organic content of material deposited on the marsh edge. The percentage of organic matter that eroded and redeposited is a highly dynamic variable that depends on the interactions between bay resuspension, external sediment supply, and marsh organic matter content.

Added the following:

“As the marsh edge erodes, the mass of mineral and organic material from each soil cohort is summed to determine the amount of mineral and organic material that is transported to the bay and therefore available for redeposition on the marsh. The organic content of the bay sediment evolves dynamically as the sum of the organic content of the eroded marsh edge sediment plus the organic content of the external sediment supply, minus the amount of organic material delivered to the marsh. Thus, the total amount of organic material deposited on the marsh depends on the organic content of the bay.”

Reviewer #2 (Remarks to the Author):

The ms is not ready for reviewing. The modeling relies on many parameters, some are empirical, some are derived from observations. Each of them has error bars. The model predictions thus also have error bars. We need to know these error bars in the values of the parameters and in the model output before we can review this manuscript in order to judge the reliability of the model predictions.

We agree with the reviewer that the model includes a number of parameters, each with their own sources of error. However, the goal of our model is to explore general behavior that emerges from novel connections between ecosystems, rather than to make precise, quantitative predictions. Below we address a) the goals of exploratory models, b) the source of parameter values that have not been reported elsewhere (i.e. the new forest carbon component of our model), and c) a full sensitivity analysis that relates model behavior to the two most influential parameter values (suspended sediment concentration and tidal range). Together, we hope these responses will allow the reviewer to better assess our work and its implications.

The changes to the text and sensitivity analysis clarify the goals of the paper (to identify underlying processes that connect carbon dynamics and geomorphology in the coastal zone) and allow the readers to evaluate potential uncertainty in the results. While quantitative results change with particular parameter values, the general behaviors are consistent across a wide parameter space (see section below on sensitivity analysis). The insights here are the conceptual patterns (not the quantitative predictions), which is consistent with the goals of exploratory models (see section below on exploratory models). The main insights from this study are:

- Climate change strengthens connectivity between coastal ecosystems, but comes with tradeoffs that include changes in where carbon is stored in the landscape.
- Carbon storage in marshes is maximized through synchronous peaks in CAR and marsh extent, which occur at intermediate rates of SLR. This pattern occurs across a range of SSC and tidal range.

Exploratory Models

We frame the model presented here as an exploratory model. We seek to highlight key feedbacks and interactions that drive the patterns of landscape change and carbon storage along the coastal transect, and do not aim to give precise quantitative predictions; this is in keeping with many geomorphic modeling frameworks (see Murray 2007 and Murray 2003 for a review of reduced complexity models in coastal geomorphology). Our goal in the development of this model was to understand basic interactions between eco-geomorphology and carbon processes, and explore potential feedbacks on a decadal-centennial time scale. Given these goals, we developed a reduced complexity model where we used simplified representations of small-scale processes to understand large-scale behavior (i.e. the emergent patterns). Therefore, interpretation of this type of model should not be based on strictly the quantitative output, but rather the trends and patterns.

Exploratory models have been used for decades in geomorphic modeling and in systems that have complex interactions across spatial and temporal scales (Larsen et al. 2014). In coastal geomorphology, they have been applied to many systems including rocky shores (Matsumoto et al. 2016, Coco et al. 2007), sandy shorelines (Ashton et al. 2001, Ashton and Murray 2006), and coastal wetlands (Mariotti and Carr 2014, Kirwan et al. 2016). Because these systems have nonlinear interactions, exploratory models can be applied to understand the underlying processes that drive change in the system and can parameterize many processes. Furthermore, these models have the benefit of reduced computing time (compared to completely detailed predictive models) and, therefore, the ability to run the models on large spatial and temporal scales. Additionally precise predictive models are typically limited to fewer processes due to computing needs. Although we simplify many processes, we are able to model the essence of many processes. In our model, the processes included are: biomass production, decomposition, wave generation, marsh edge erosion, marsh vertical accretion, upland marsh migration, and forest soil development. These processes were selected as those most likely to be first-order controls on the coastal zone evolution and carbon storage.

We added the following to the text in the model approach section about exploratory models:

“This type of exploratory model has been used for decades in geomorphology as a way to identify and understand feedbacks in nonlinear systems^{34,35}.”

Parameterization of the forest-carbon model component

Nearly all parameters in the model have been previously published in numerical models of marsh evolution and the sensitivity of these parameters tested within those publications (sources listed in Supplemental Table 1 in the text). Those that are empirical (i.e. derived from observations) are limited to the forest carbon model, which was designed based on field and remote sensing data from Smith and Kirwan 2021 and Chen and Kirwan, accepted in principle at Nature Geoscience (data provided as Supplementary Fig 1. for this case). We clarified the source of the data for the parameters in Supplementary Table 1. While both forms of data have error bars on them (added to Supplementary Fig. 1), both datasets indicate the same trends in forest biomass. For example, Smith and Kirwan (2021) observed living tree biomass to increase with distance from the marsh forest boundary, while total soil carbon was greatest near the marsh forest boundary (Figure R2). Chen and Kirwan (accepted in principle), also observed an increase in aboveground carbon with distance from the marsh-forest boundary (Figure R3). The combination of these two types of data led to the development of the equation to describe both soil carbon and biomass at the marsh-forest transition. The data were used in the development of the form of the equation and not directly implemented in the model.

Figure R2. Figure from Smith and Kirwan 2021 demonstrating carbon stocks at the marsh-forest transition.

Figure R3. Reiteration of Supplementary Fig. 1, which demonstrates the remote sensing data used to construct the model of forest biomass along the marsh-forest transition. Data from Chen and Kirwan (accepted in principle).

Sensitivity Analyses

You make an excellent point that there is a lot of uncertainty in our results, and have responded by adding a full sensitivity analysis on the parameters most likely to influence model behavior. We ran over 300 additional model runs to test how sediment supply and tidal range influence our results, similar to sensitivity analysis done in other exploratory models (Mariotti and Carr 2014; Kirwan et al. 2016). Reviewer 1 identified SSC input as a potentially important/sensitive variable. We tested the model output for a range of SSC values (10-100 mg/L) to determine the relative importance of this parameter (Supplementary Fig. 2, attached below). The following text was added to the supplemental information:

“To test model sensitivity to key environmental parameters driving marsh evolution, we ran the model for a wide range of input suspended sediment concentrations and tidal ranges. Sediment supply is a key driver of the evolution of natural marshes^{74,75}, and the suspended sediment concentration (SSC) of the bay is a key parameter affecting marsh sustainability in numerical models^{76,77}. Under low to moderate rates of SLR [1-6 mm yr⁻¹], SSC input did not substantially change any of the key model results we explored (Supplementary Fig. 2). At higher rates of SLR [>6 mm yr⁻¹] and intermediate SSC [20-90 mg L⁻¹], we also observed no substantial changes in the metrics used in our study. However, at both high rates of SLR and extremely small or large SSC, the model results are substantially impacted. At low SSC [10 mg L⁻¹], marsh width, CAR, and marsh C are reduced, while at high rates of SLR and SSC = 100 mg L⁻¹, marsh width is increased. Forest C is not affected by changes in SSC. These results indicate that the model is not overly sensitive to SSC, and that consistent results are obtained over a wide range of reasonable SSC [i.e. 20-90 mg L⁻¹]. Within this range of SSC [20-90 mg L⁻¹], we further observe synchronous peaks in marsh width, CAR, and autochthonous CAR at intermediate rates of SLR (Supplementary Fig. 3). Together, this sensitivity analysis highlights that landscape carbon is driven mechanistically by synchronous peaks in CAR and marsh width, independent of the external sediment supply.”

Supplementary Figure 2. Relative change in marsh width (a), carbon accumulation rate (b), total marsh carbon (c), and total forest carbon (d) for a range of suspended sediment concentrations and SLR rates. Relative change was calculated as the difference between model runs presented in the main text (Figs. 3a-4, SSC=50 mg L⁻¹, SLR = 1-15 mm yr⁻¹) and model runs presented here (SSC=10-100 mg L⁻¹, SLR = 1-15 mm yr⁻¹). The vertical box in each figure indicates the model runs in the main text.

Supplementary Figure 3. CAR and marsh width for $SSC=20 \text{ mg L}^{-1}$ (a) and $SSC=90 \text{ mg L}^{-1}$ (b). While the values of CAR and marsh width change depending on the suspended sediment concentration, the general patterns observed remain consistent, with a peak in all metrics (except allochthonous C) at intermediate rates of SLR.

We also tested the model sensitivity to tidal range and added the following text and figure to the supplemental material:

“Another dominant factor in marsh evolution is tidal range^{76,77}. To test the sensitivity to tidal range, we ran the model using two tidal ranges (1 and 3 m), in addition to the 1.4 m tidal range considered in the simulations presented in the main text. These results demonstrate that allochthonous carbon deposition increases with tidal range (Supplementary Fig. 4). Like the original simulations, marsh width and CAR tend to increase with SLR towards an optimum SLR rate regardless of tidal range. However, with a large tidal range, only the rising limb of marsh extent and CAR are observed (Supplementary Fig. 4). We attribute this pattern to the lack of marsh drowning in the experiment, driven by spinup conditions that created a marsh that was initially higher in elevation, and the well-known link between tidal range and marsh sustainability^{76,77}.”

Supplementary Figure 4. CAR and marsh width for tidal range = 1 m (a) and tidal range = 3 m (b). With a small tidal range, CAR and marsh width peak synchronously at an intermediate rate of SLR. At a higher tidal range, all metrics continue to increase with SLR because the marsh platform started at a higher elevation and drowning did not occur.

Finally, we have added envelopes to model results in Fig. 2 to illustrate how suspended sediment supply influences the exchange of carbon between the marsh and bay (see revised Fig. 2 below). The addition of the envelopes clearly demonstrates the level of uncertainty (or variability) of our results depending on parameter choice, but that the basic behavior of the system remains consistent. Adding envelopes to Fig. 3 made it difficult to read. Instead we included Supplementary Fig. 3, which shows the bounds of the envelopes for Fig. 3 if one considers a range of sediment supply.

Figure 2. (A) Fluxes (kg/yr per meter of marsh edge) of carbon eroded from the marsh edge into the bay (blue line) and of allochthonous carbon deposited on the marsh platform from the bay (red line). The carbon flux from the marsh to the bay represents the mass flux of carbon eroded from the marsh edge, averaged over the last 50 years of the model experiment. The carbon flux from the bay to the marsh represents the mass flux of allochthonous carbon deposited on the marsh surface (surface deposition), averaged over the last 50 years of the model experiment. (B) The material eroded from the marsh edge makes up ~15% of the total carbon deposited on the marsh surface. This contribution decreases until an intermediate rate of SLR, and then increases. Colored envelopes show the variation (minimum and maximum) in results with $\text{SSC}=20\text{-}60$ mg/L .

We also tested model sensitivity to external sediment supply in the absence of allochthonous carbon deposition. When all allochthonous carbon is completely mineralized, the results (marsh extent, carbon) are more sensitive to external sediment supply (Fig. R4). The presence of allochthonous carbon is a buffer for marshes, and allows the basic underlying patterns of CAR to persist under a wide range of conditions. Without allochthonous carbon, the CAR patterns are strongly affected by low sediment supply ($10\text{-}30$ mg L^{-1}) across the entire range of SLR tested, indicating that without external sediment supply and allochthonous C delivery, the patterns of marsh carbon accumulation are altered. Namely, CAR and marsh C are higher at low SSC and SLR, relative to the model results presented in the paper, driven by

increases in autochthonous production. However, marsh width and total forest carbon are not substantially affected by changes in external sediment supply (Fig. R4a and R4d). Marsh width is only affected at high rates of SLR and extremely large or small sediment supplies [10 or 100 mg L⁻¹], while patterns in forest carbon are insensitive to these variables. These analyses confirm a consistent peak in marsh width and CAR at intermediate rates of SLR, and demonstrate the importance of allochthonous C.

Figure R4. Relative change in marsh width (a), carbon accumulation rate (b), total marsh carbon (c), and total forest carbon (d) for a range of suspended sediment concentrations and SLR rates in a scenario with no allochthonous carbon deposition. Relative change was calculated as the difference between model runs presented in the main text (Figs. 3a-4, SSC=50 mg L⁻¹, SLR = 1-15 mm yr⁻¹) and model runs presented here (SSC=10-100 mg L⁻¹, SLR = 1-15 mm yr⁻¹). The vertical box in each figure indicates the model runs in the main text.

Additional References in Response to Reviewers:

Ashton, A., Murray, A. B., & Arnoult, O. (2001). Formation of coastline features by large-scale instabilities induced by high-angle waves. *Nature*, 414(6861), 296-300.

Ashton, A. D., & Murray, A. B. (2006). High-angle wave instability and emergent shoreline shapes: 1. Modeling of sand waves, flying spits, and capes. *Journal of Geophysical Research: Earth Surface*, 111(F4).

Coco, G., Murray, A. B., & Green, M. O. (2007). Sorted bed forms as self-organized patterns: 1. Model development. *Journal of Geophysical Research: Earth Surface*, 112(F3).

Larsen, L., Thomas, C., Eppinga, M., & Coulthard, T. (2014). Exploratory modeling: Extracting causality from complexity. *Eos, Transactions American Geophysical Union*, 95(32), 285-286.

Matsumoto, H., Dickson, M. E., & Kench, P. S. (2016). An exploratory numerical model of rocky shore profile evolution. *Geomorphology*, 268, 98-109.

Murray, A. B. (2003). Contrasting the goals, strategies, and predictions associated with simplified numerical models and detailed simulations. *Geophysical Monograph-American Geophysical Union*, 135, 151-168.

Reviewer #1 (Remarks to the Author):

Dear Authors,

Thank you for thoroughly addressing my comments and concerns. I greatly appreciate your detailed responses. The added text and simulations associated with the sensitivity analysis have answered my major comments and questions and improved the manuscript.

I have a few minor comments outlined below.

Sensitivity analysis – Did you consider varying upland slope which will impact marsh migration and landscape size?

Line 115 – Please add reference for field measurements.

Figure 4 – I suggest explicitly stating in the caption that the dots are SLR rates increasing consecutively.

Line 204-207 –Would erosion not be considered disruption? This statement would only hold under the assumption that allochthonous carbon from edge erosion and deposition undergoes less disturbance than if remained intact (or other future condition). Perhaps rephrase for clarity or expand discussion.

Line 209 – This is interesting. I suggest calculating this oversight to increase the impact of this statement.

Line 224 – I suggest adding a sentence here about system understanding and applications to different coastal systems. I recognize that this model is exploratory and does not represent a specific coastline, but it would be beneficial to add some text reflecting on different marsh systems and variability in responses based on these findings.

Line 237-238 – I suggest expanding on this point by more explicitly explaining how these model results explain contradictory results observed in marshes?

Line 341 – I suggest adding a reference to this sentence.

Reviewer #2 (Remarks to the Author):

It is very hard to review this paper because doing hundreds of numerical simulations produces a range of output but we still don't know which output is the most reliable one, especially that it is not just a matter of changing one parameter at a time but it may require changing two or more parameters at the same time. The modeling is qualitative and not quantitative. Sorry. The results are interesting but I do not think that they are convincing as the modeling is qualitative.

Reviewer #3 (Remarks to the Author):

This is a review for the manuscript entitled "Climate-driven tradeoffs between landscape connectivity, ecosystem extent, and the maintenance of the coastal carbon sink" submitted by Valentine and co-authors to Nature Communications. In general, my comments are relatively minor, and I find the modeling approach to be cohesive and well-explained, and the interpretation of results to be both appropriate and relevant. I do think that more care is needed to contextualize how model decisions and assumptions may bias interpretations, and more effort is needed to make this study accessible to a broader leadership (see general comments below). Overall, I recommend some minor revisions are needed prior to publication.

General comments

• **Intro:** since the focus of this study is models, it would be good to introduce the current status and potential growth areas for coastal carbon models to better contextualize the need for (and scope of) this work, especially since the reader likely goes from intro to

results/discussion.

- While I don't have issues with the model parameterization decisions, they differ from the reality of carbon lability, and addressing the potential bias of the model outcomes more consistently throughout the main text is important for reader context.
- The manuscript is skewed more towards results than discussion: interpretation of what your results mean would be useful, especially considering how your model results translate to other systems. Since the readership of this journal is large, considering other common forest-marsh-sediment topologies and how your results may differ in those systems would make your story more impactful to a wider swath of the broad readership of this journal.

Specific comments

- L45-46: this is a bit of a jump, could use a better ideological segway between these thoughts
- L65: suggest removing uniquely
- L69-71: This language makes it sound like these are the sole factors that influence geomorphology, rather than the dominant drivers / selected model parameters. Consider revising for clarity.
- L82+: Would be good to mention very briefly what constraints on autochthonous production are considered (i.e., assumptions about vegetation community and climate)
- L124-127: I'm unclear how bulk results support multiple specific mechanisms. Can you add additional clarity here (i.e., results match hypotheses based on each of the listed mechanisms)? Some more details explaining this statement would be helpful.
- Figure 1: Using allochthonous instead of total C is a little confusing here: although autochthonous inputs are shown in arrows, they are not included. Is the idea that autochthonous material is mineralized/otherwise consumed or transported off the marsh? The autochthonous arrow in the marsh seems to indicate burial, and some of that carbon is surely either already, or will be transformed into recalcitrant material. Alternatively, is this figure only sediment accretion, that could be clarified a bit more / panel A doesn't necessarily fit (could put the full panel A into supplemental, and have only the relevant mechanisms in the new panel A).
- L133: Is forest retreat represented in model runs by the interface between allochthonous C and forest soil at the surface? If so, it would be helpful to move the dashed line representing that interface to the appropriate spot in each panel to show forest retreat.
- L137: see Figure 1 comment for clarifying autochthonous content representation
- L159: based on your discussion of recycled carbon: that would lead to exposure of anoxically stored carbon, which y'all assume is refractory, to oxic seawater during erosion, then sunlight, heat, and oxygen after deposition. With the assumption that some biorefractory material is photolabile and vice versa, this seems like respiration may be a significant, but unaccounted-for component of these processes. It could be really interesting to explore your results in the context that OM has a gradient of labilities that shifts under different environmental conditions.
- L190: up "to" half
- L220+: What is the feedback in your models that drives lower autochthonous carbon at SLR > 8mm/yr in Fig 3B relative to 3A? Does this suggest that allochthonous carbon, by maintaining accretion rates, promotes autochthonous production under higher SLR scenarios? Would be good to discuss this component of the autochthonous trend, which contrasts with "higher autochthonous C without allochthonous" at lower SLR rates.
- L230: Looking at Figure 4A, it looks like the carbon stock maximum is 10 or 11 mm/yr, and decreases slower at >10/11 mm/yr than it increases at < 10/11 mm/yr. Double-check interpretation.
- Figure 4B: I'm curious why panel B doesn't include a landscape line, which is helpful for interpreting net trends in 4A. Also it would be good to clarify how ecosystems are weighted (L393 says they are very different widths, so I assume different areas). % change may be a better way to apples-to-apples your inter-ecosystem comparisons (and more applicable across other systems where areal proportions differ widely from your model system).
- Figure 4C: this is only referenced once that I find in the manuscript, and differs in

structure and message from A and B. I suggest it should be a stand-alone supplemental figure, to make Figure 4 simpler and more cohesive.

- **L234: extra space: "cycling , "**

- **L233: is increased accumulation space a result of eroding the marsh edge? Would be good to clarify that**

- **L337: this keeps the model much simpler, but, per comments above, some portion will likely be labile, and I think highlighting this assumption and the associated error in the main text is important to properly contextualize your results.**

- **L345: I'm curious about setting biomass to be maximized at an intermediate elevation. Is this based on vegetation assessment? If so, please cite, if not, I'm curious what impact that might have on the results.**

Reviewer #1 (Remarks to the Author):

Dear Authors,

Thank you for thoroughly addressing my comments and concerns. I greatly appreciate your detailed responses. The added text and simulations associated with the sensitivity analysis have answered my major comments and questions and improved the manuscript.

I have a few minor comments outlined below.

Thank you for the additional comments. We have addressed them below, and we paid particular attention to the role of C decomposition, as this was brought up by both you and Reviewer #3.

Sensitivity analysis – Did you consider varying upland slope which will impact marsh migration and landscape size?

Upland slope is one of the two drivers of marsh migration within the model and therefore has a substantial impact on the outcome of marsh and forest extent. The effect of slope on marsh and forest extent was fully explored in an earlier version of the model presented here in Kirwan et al. 2016. They tested a suite of upland slopes, ranging from a gentle sloping coastal plain (Slope = 0.001) to a high slope representing an active margin coast (Slope = 0.2) (Figure R1). They found that at high rates of SLR, the marsh width declined rapidly, representing marsh drowning – regardless of upland slope. However, upland slope did have a larger impact on marsh extent with lower rates of SLR; with lower SLR, the marsh was able to expand due to marsh migration with low upland slopes, but not with high upland slopes (where marsh migration cannot occur).

Redacted

Figure R1. Reproduction of Figure 3 and caption from Kirwan et al. 2016 demonstrating different behaviors with changing upland slope.

For this manuscript, we were primarily interested in when the adjacent ecosystems interact, and therefore chose a low slope value where marsh migration is possible. If we chose a high slope (e.g., 0.2), marsh migration would not occur, and the forest ecosystem would remain the same throughout the simulation, while the marsh will only erode. To demonstrate this change, we reanalyzed the data from the existing model runs but excluded any marsh migration (i.e. cut off marsh extent at the original marsh-forest boundary). With no marsh migration, the bay dynamics remain the same as with marsh migration, forest size

does not change with SLR, and marshes have net loss in area compared to net gain for all rates of SLR (Figure R2).

Figure R2. Change in marsh ecosystem size (a) with marsh migration and (b) without marsh migration. Note different y-axis scales.

We added this to the discussion of other marsh-forest topologies:

“The importance of carbon connectivity highlighted on this generalized coastline can be extrapolated to other coastal marsh systems. For example, connectivity may be reduced in areas where marsh migration is hindered (urban development, steep upland slope). This restriction may result in decreased marsh extent and marsh carbon storage. Similarly, increased erosion of the marsh edge (high winds, more exposed coastline) would increase the exchange of carbon across the bay-marsh interface. The increased connectivity from edge erosion would increase suspended sediment and allochthonous carbon adjacent to the marsh, resulting in higher CAR and enhanced marsh resilience to SLR. Likewise, increased tidal range increases the connectivity between the marsh and the bay, resulting in higher CAR and marsh extent (Supplementary Fig. 5). Our results underline the importance of connectivity for increased coastal resilience and carbon storage. While this model describes qualitative patterns in coastal landscape response to global change, it highlights the need for more robust couplings between interacting habitats in earth system models as we demonstrate these couplings fundamentally alter landscape carbon balances^{27,72}.”

Line 115 – Please add reference for field measurements.

Added reference to Smith and Kirwan, 2021

“Therefore, the model considers a single value for net carbon accumulation, based on field measurements (Smith and Kirwan, 2021), that reflects both deposition and decomposition...”

Figure 4 – I suggest explicitly stating in the caption that the dots are SLR rates increasing consecutively.

Changed to include (now Supplementary Figure 6):

“Each dot represents one model run with incremental increases in SLR rate.”

Line 204-207 –Would erosion not be considered disruption? This statement would only hold under the assumption that allochthonous carbon from edge erosion and deposition undergoes less disturbance than if remained intact (or other future condition). Perhaps rephrase for clarity or expand discussion.

This is an important point, and a great point for further model development and investigation. Reviewer 3 asked a number of questions regarding lability of carbon and about the erosion of autochthonous material that becomes allochthonous (and therefore recalcitrant), so please see response to General Comment #2 from reviewer 3 for a more complete answer. In short, erosion would be considered a disruption and likely material released from marsh edge erosion would be decomposed once exposed to the water column. However, the contribution of carbon from the marsh edge is small (~15% of allochthonous C, Figure 2 in the main text), and the majority of allochthonous C comes from the external sediment supply and the bay bottom, where carbon is tightly bound to sediment particles and would decompose very little. (See below response to Reviewer 3 for a more complete response.)

The following was added to the text for clarification and discussion of the assumptions about carbon lability:

We added the following to the methods:

“This assumption potentially overestimates the amount of organic matter in the system, as marsh erosion exposes and disturbs previously-buried carbon. However, marsh edge erosion makes up a small component of total allochthonous carbon (15%, Fig. 2b); the remaining allochthonous carbon is from the external sediment supply and resuspension of the bay bottom. These sources are repeatedly disturbed and any remaining carbon is recalcitrant and tightly bound to sediment⁴¹.”

We added the following paragraph to the discussion:

“Furthermore, these model experiments give insight into our parameterization of organic matter decomposition and carbon lability. While in the first set of experiments all allochthonous carbon is refractory, these later experiments parametrize all allochthonous marsh carbon as labile with a very high decomposition rate (100% decomposes instantaneously). While the total amount of marsh carbon is sensitive to the amount of recalcitrant allochthonous carbon (indicated by differences between Fig. 3a and Fig. 3b), marsh extent and carbon storage peak at intermediate rates of SLR independent of the lability of allochthonous carbon. This highlights the underlying behavior of marshes and their ability to adapt to changing sea levels, independent of carbon lability parameterizations. However, the differences in the SLR tipping point and total amount carbon demonstrate the need to better understand carbon lability in coastal systems.”

Line 209 – This is interesting. I suggest calculating this oversight to increase the impact of this statement.

Great idea. Added the following:

“In the scenario presented here, allochthonous carbon contribution to marsh accretion quadruples as SLR increases from 2 mm yr⁻¹ to 7-15 mm yr⁻¹ (Fig. 2).”

Line 224 – I suggest adding a sentence here about system understanding and applications to different coastal systems. I recognize that this model is exploratory and does not represent a specific coastline, but it would be beneficial to add some text reflecting on different marsh systems and variability in responses based on these findings.

We added the following (later in the text than you suggested because of other added text):

“The importance of carbon connectivity highlighted on this generalized coastline can be extrapolated to other coastal marsh systems. For example, connectivity may be reduced in areas where marsh migration is hindered (urban development, steep upland slope). This restriction may result in decreased marsh extent and marsh carbon storage. Similarly, increased erosion of the marsh edge (high winds, more exposed coastline) would increase the exchange of carbon across the bay-marsh interface. The increased connectivity from edge erosion would increase suspended sediment and allochthonous carbon adjacent to the marsh, resulting in higher CAR and enhanced marsh resilience to SLR. Likewise, increased tidal range increases the connectivity between the marsh and the bay, resulting in higher CAR and marsh extent (Supplementary Fig. 5). Our results underline the importance of connectivity for increased coastal resilience and carbon storage. While this model describes qualitative patterns in coastal landscape response to global change, it highlights

the need for more robust couplings between interacting habitats in earth system models as we demonstrate these couplings fundamentally alter landscape carbon balances^{27,72}."

Line 237-238 – I suggest expanding on this point by more explicitly explaining how these model results explain contradictory results observed in marshes?

This sentence was expanded to make the meaning clearer:

"This nonlinear response may help explain the seemingly contradictory results observed in marshes, where both positive and negative relationships between SLR and CAR have been observed ^{61,65}"

Line 341 – I suggest adding a reference to this sentence.

Added Van de Broek et al., 2018 citation:

"The carbon is distributed equally across the bay bottom, and once it enters the bay is considered allochthonous and therefore does not decompose (Van de Broek et al., 2018)."

Reviewer #2 (Remarks to the Author):

It is very hard to review this paper because doing hundreds of numerical simulations produces a range of output but we still don't know which output is the most reliable one, especially that it is not just a matter of changing one parameter at a time but it may require changing two or more parameters at the same time. The modeling is qualitative and not quantitative. Sorry. The results are interesting but I do not think that they are convincing as the modeling is qualitative.

Reviewer #3 (Remarks to the Author):

This is a review for the manuscript entitled "Climate-driven tradeoffs between landscape connectivity, ecosystem extent, and the maintenance of the coastal carbon sink" submitted by Valentine and co-authors to Nature Communications. In general, my comments are relatively minor, and I find the modeling approach to be cohesive and well-explained, and the interpretation of results to be both appropriate and relevant. I do think that more care is needed to contextualize how model decisions and assumptions may bias interpretations, and more effort is needed to make this study accessible to a broader leadership (see

general comments below). Overall, I recommend some minor revisions are needed prior to publication.

General comments

- Intro: since the focus of this study is models, it would be good to introduce the current status and potential growth areas for coastal carbon models to better contextualize the need for (and scope of) this work, especially since the reader likely goes from intro to results/discussion.

We agree that it is a good idea to contextualize our model with other coastal models. We found that adding statement in the introduction were redundant with the very beginning of the "Model approach and basic behavior" immediately after the introduction (lines 64-67) and the description of blue carbon science (starting on line 35). Instead of adding material to the introduction, we added the following to the discussion:

"Our results underline the importance of connectivity for increased coastal resilience and carbon storage. While this model describes qualitative patterns in coastal landscape response to global change, it highlights the need for more robust couplings between interacting habitats in earth system models as we demonstrate these couplings fundamentally alter landscape carbon balances^{27,72}."

We also clarified in the introduction:

"Given these contrasting responses, the net impacts of SLR on coastwide carbon remain largely unknown and is not explicitly included in models of coastal carbon dynamics."

- While I don't have issues with the model parameterization decisions, they differ from the reality of carbon lability, and addressing the potential bias of the model outcomes more consistently throughout the main text is important for reader context.

You bring up a great point about lability of carbon. As you guessed, this was a simplification in the parameterization and does add some error to our results and is worth exploring in the manuscript. The model breaks carbon into two fractions – labile and refractory. In the model, all allochthonous material is considered refractory, while all autochthonous material is labile.

Autochthonous carbon includes lignin and other more refractory components, indicating that not all of the carbon from this fraction is labile (Fig. R3a). However, we model all autochthonous C as labile (Fig. R3b). While this simplifies the model and removes complexity and does not account for all processes, we reach a similar finding in the total amount of carbon at depth (0.4 m, dashed line in Fig. R3). The labile autochthonous carbon

decomposes as a function of burial depth, and ceases to decompose beyond a sediment depth of 0.4 m. In the case where autochthonous C is both labile and refractory, the refractory component remains constant with depth and the labile component decomposes with depth (Fig. R3a). Since the amount of labile material is less in this case compared to when all autochthonous C is labile, less of it decomposes (given the form of the equation). When all the material is labile, more of it decomposes since the initial value is higher. In essence the amount of remaining "labile" material at depth would actually be composed of primarily the refractory material.

Figure R3. Labile and refractory carbon with depth as (a) observed in the field and (b) represented in our model framework.

Given this parameterization and the fact that even some "refractory" material may decompose when disturbed (or subjected to light), we likely overestimate the carbon that is eroded from the marsh edge and remains in the system. However, the rate of marsh edge

erosion is small. For example, as demonstrated in Figure 2B, carbon from marsh edge erosion make up about 15% of the allochthonous C across a range of sea level rise rates and suspended sediment concentrations. The rest of this carbon comes from the external sediment supply and bay bottom. The sediments in these areas are likely remobilized frequently from waves and currents and are likely to be mostly recalcitrant and have little labile material remaining or are tightly bound to sediment particles.

We can further explore this by considering the two end member cases, (1) where all allochthonous carbon does not decompose and (2) complete mineralization of allochthonous carbon. This is presented in the manuscript in Figures 3a and 3b, respectively. The differences between these two cases represent the model's sensitivity to allochthonous carbon decomposition. The greater the preservation of allochthonous carbon (i.e. refractory carbon), the greater the marsh extent and greater storage of carbon in the marsh itself. Any partial decomposition of allochthonous material (from disturbance, light/heat, oxygenation biodecomposition, etc) would give results with lower total carbon values and lower marsh extents compared to our findings (Figure 3a), but higher than the case where all allochthonous material is decomposed (Figure 3b). While the findings are more striking when allochthonous carbon is preserved entirely, the same trends in marsh extent and marsh carbon exist with no allochthonous carbon, demonstrating that the model is capturing the processes that drive marsh (and carbon) evolution, even though the quantitative results depend on how decomposition of allochthonous material is parameterized. You make a further interesting observation that the decomposition may change over time as well. For example, with climate change we may expect higher temperatures and higher decomposition rates of refractory carbon, so at high rates of SLR (presumably higher temperatures), decomposition may be higher compared to decomposition at lower rates of SLR, thus altering the shape of the relationship between SLR and CAR.

It is worth noting that in most landscape-scale geomorphic models, carbon is parameterized even more simply than in our model (e.g., Mariotti and Carr, 2014; Kirwan et al., 2016, Carr et al., 2020) and that we provide a step forward in coupled geomorphic-carbon modeling by allowing for two organic carbon fractions and tracking these components.

We added the following to the methods:

"This assumption potentially overestimates the amount of organic matter in the system, as marsh erosion exposes and disturbs previously-buried carbon. However, marsh edge erosion makes up a small component of total allochthonous carbon (15%, Fig. 2b); the remaining allochthonous carbon is from the external sediment supply and resuspension of the bay bottom. These sources are repeatedly disturbed and any remaining carbon is recalcitrant and tightly bound to sediment⁴¹."

We added the following paragraph to the discussion:

“Furthermore, these model experiments give insight into our parameterization of organic matter decomposition and carbon lability. While in the first set of experiments all allochthonous carbon is refractory, these later experiments parametrize all allochthonous marsh carbon as labile with a very high decomposition rate (100% decomposes instantaneously). While the total amount of marsh carbon is sensitive to the amount of recalcitrant allochthonous carbon (indicated by differences between Fig. 3a and Fig. 3b), marsh extent and carbon storage peak at intermediate rates of SLR independent of the lability of allochthonous carbon. This highlights the underlying behavior of marshes and their ability to adapt to changing sea levels, independent of carbon lability parameterizations. However, the differences in the SLR tipping point and total amount carbon demonstrate the need to better understand carbon lability in coastal systems. ”

- The manuscript is skewed more towards results than discussion: interpretation of what your results mean would be useful, especially considering how your model results translate to other systems. Since the readership of this journal is large, considering other common forest-marsh-sediment topologies and how your results may differ in those systems would make your story more impactful to a wider swath of the broad readership of this journal.

Great idea. We added the following to the manuscript:

“The importance of carbon connectivity highlighted on this generalized coastline can be extrapolated to other coastal marsh systems. For example, connectivity may be reduced in areas where marsh migration is hindered (urban development, steep upland slope). This restriction may result in decreased marsh extent and marsh carbon storage. Similarly, increased erosion of the marsh edge (high winds, more exposed coastline) would increase the exchange of carbon across the bay-marsh interface. The increased connectivity from edge erosion would increase suspended sediment and allochthonous carbon adjacent to the marsh, resulting in higher CAR and enhanced marsh resilience to SLR. Likewise, increased tidal range increases the connectivity between the marsh and the bay, resulting in higher CAR and marsh extent (Supplementary Fig. 5). Our results underline the importance of connectivity for increased coastal resilience and carbon storage. While this model describes qualitative patterns in coastal landscape response to global change, it highlights the need for more robust couplings between interacting habitats in earth system models as we demonstrate these couplings fundamentally alter landscape carbon balances^{27,72}.”

Specific comments

- L45-46: this is a bit of a jump, could use a better ideological segway between these thoughts

Added to read:

"Therefore, it is unclear if a negative carbon-climate feedback will persist, as the fate of coastal carbon depends not only on how CAR responds to SLR, but also on how the size, configuration, and interactions of the coastal system respond."

- L65: suggest removing uniquely

Changed

- L69-71: This language makes it sound like these are the sole factors that influence geomorphology, rather than the dominant drivers / selected model parameters. Consider revising for clarity.

Added:

"...key drivers of..."

- L82+: Would be good to mention very briefly what constraints on autochthonous production are considered (i.e., assumptions about vegetation community and climate)

Later in our response, we provide more background on the parabolic relationship used to determine biomass (i.e. autochthonous production), but briefly, this relationship is commonly used to represent marsh plant biomass in respect to elevation and has been used to model ecogeomorphic evolution spanning vegetation communities and climate (Mudd et al. 2004, van de Koppel et al. 2005, D'Alpaos et al. 2007, Kirwan and Murray 2007, Kirwan et al. 2010, Larsen and Harvey 2010, Lorenzo-Trueba et al. 2010, Mariotti and Carr, 2014, Kirwan et al. 2016, Carr et al. 2018). The relationship used in this study represents a single species, *Spartina alterniflora* in a mid-latitude salt marsh. This assumption has been added to the manuscript:

"Belowground biomass, represented as a monoculture of *Spartina alterniflora*, is modeled as a parabolic function of elevation which peaks at an intermediate elevation ($B_{max} = 1000 \text{ gm}^{-2}$)^{31,36,38}."

- L124-127: I'm unclear how bulk results support multiple specific mechanisms. Can you add additional clarity here (i.e., results match hypotheses based on each of the listed mechanisms)? Some more details explaining this statement would be helpful.

Added the following text:

“As expected, the marsh sediment profile is deeper and the marsh platform is wider with moderate SLR (Fig. 1c) compared to historical SLR (Fig. 1b), reflecting increased vertical accretion rates and faster marsh migration. Correspondingly, CAR (Fig. 1b, 1c, Supplementary Fig. 2) and marsh productivity (Supplementary Fig. 1) were also higher with moderate SLR than with historical SLR.”

- Figure 1: Using allochthonous instead of total C is a little confusing here: although autochthonous inputs are shown in arrows, they are not included. Is the idea that autochthonous material is mineralized/otherwise consumed or transported off the marsh? The autochthonous arrow in the marsh seems to indicate burial, and some of that carbon is surely either already, or will be transformed into recalcitrant material. Alternatively, is this figure only sediment accretion, that could be clarified a bit more / panel A doesn't necessarily fit (could put the full panel A into supplemental, and have only the relevant mechanisms in the new panel A).

We agree that Figure 1 could use clarification, and thank you for the suggestions. Panel A is meant to serve as a schematic of the entire model system, including all processes – both carbon and geomorphic -- that go into the model. The reason for including it with panels B and C is to give context to the results and show the entire domain, especially since panels B and C only show the allochthonous C, and not all the results of the model. We opted to visualize the allochthonous C in these panels because it is the crux of the story that we are telling in the manuscript. Our main point is that allochthonous C is more important than previously thought. On the other hand, the findings of autochthonous carbon are less novel, and therefore not the focus of the manuscript. To improve the figure and/or manuscript, we:

- Changed the dashed lines running vertically to represent the different ecosystems in each panel (Fig. 1)
- Highlighted in panel A the allochthonous C processes (bolded the arrows) and added to the caption that indicates that all processes are shown in panel A, while only allochthonous C is shown in the later panels
- Clarified in the caption that only allochthonous carbon was shown in panels B and C
- Added and cited supplementary figures that show autochthonous C and total C (Supplementary Fig. 1, Supplementary Fig. 2)

The figure and caption now read:

Figure 1. Schematic of 2D transect model of the bay-marsh-forest system representing all modeled processes (a). Geomorphic processes are indicated with black arrows, while carbon processes are in green (autochthonous, Supplementary Fig. 1) and white (allocthonous, shown in panels b and c). The coastal transect was subjected to low (b) and moderate (c) rates of sea level rise (SLR), which resulted in more allocthonous carbon under high rates of SLR. Model experiments were conducted under a 50 mg L^{-1} sediment supply and a 1.4 m tidal range. Color shadings along scale on right indicate the amount of allocthonous carbon [g]. Underlying stratigraphy was generated during the model spinup. x-axis distance is relative to initial shoreline position and y-axis is relative to initial sea level. Vertical dashed lines delineate bay-marsh and marsh-forest boundary positions at the end of the model simulations. Total carbon is presented in Supplementary Fig. 5.

The Supplementary Information now include the following:

Supplementary Figure 1. Autochthonous carbon after the coastal transect was subjected to low (a) and moderate (b) rates of sea level rise (SLR). Model experiments were conducted under a 50 mg L⁻¹ sediment supply and a 1.4 m tidal range. Color shadings along scale on right indicate the amount of autochthonous carbon [g]. Underlying stratigraphy was generated during the model spinup. x-axis distance is relative to initial shoreline position and y-axis is relative to initial sea level.

Supplementary Figure 2. Total Carbon after the coastal transect was subjected to low (a) and moderate (b) rates of sea level rise (SLR). Model experiments were conducted under a 50 mg L⁻¹ sediment supply and a 1.4 m tidal range. Color shadings along scale on right indicate the total amount of carbon [g]. Underlying stratigraphy was generated during the model spinup. x-axis distance is relative to initial shoreline position and y-axis is relative to initial sea level.

- L133: Is forest retreat represented in model runs by the interface between allochthonous C and forest soil at the surface? If so, it would be helpful to move the dashed line representing that interface to the appropriate spot in each panel to show forest retreat.

Great point – yes it is! Changed and included in response to the previous comment regarding Figure 1.

- L137: see Figure 1 comment for clarifying autochthonous content representation

See above response for more on Figure 1. We clarified in the caption that only allochthonous carbon is represented, and then added supplemental figures with autochthonous C and total C.

• L159: based on your discussion of recycled carbon: that would lead to exposure of anoxically stored carbon, which y'all assume is refractory, to oxic seawater during erosion, then sunlight, heat, and oxygen after deposition. With the assumption that some biorefractory material is photolabile and vice versa, this seems like respiration may be a significant, but unaccounted-for component of these processes. It could be really interesting to explore your results in the context that OM has a gradient of labilities that shifts under different environmental conditions.

Thank you for this comment – it merits further explanation in the manuscript. See above response to General Comment 2.

• L190: up “to” half

Changed

• L220+: What is the feedback in your models that drives lower autochthonous carbon at SLR > 8mm/yr in Fig 3B relative to 3A? Does this suggest that allochthonous carbon, by maintaining accretion rates, promotes autochthonous production under higher SLR scenarios? Would be good to discuss this component of the autochthonous trend, which contrasts with “higher autochthonous C without allochthonous” at lower SLR rates.

Yes – exactly! Increased allochthonous inputs allow the marsh to maintain elevation (i.e. position in the tidal frame), maximizing autochthonous production (Figure R4). If the elevation declines beyond a certain point – which occurs at high rates of sea level rise -- plants are less productive and eventually this reduction in autochthonous production leads to largescale marsh drowning.

Figure R4. Schematic of biomass – elevation relationship. When the marsh is high in the tidal frame ($t=1$), biomass is relatively low. As the marsh loses elevation due to SLR ($t=2$), biomass increases. However, if the marsh becomes too low in the tidal frame ($t=3$), biomass decreases. The presence of allochthonous material allows the marsh to maintain a position where biomass is increased compared to a marsh that is high in the tidal frame.

We added the following to the manuscript:

“However, at high rates of SLR [$>8 \text{ mm yr}^{-1}$] autochthonous carbon cannot compensate for the lack of allochthonous carbon. At these high rates of SLR, elevations have decreased so as to lead to decreased plant production and less autochthonous accretion.”

- L230: Looking at Figure 4A, it looks like the carbon stock maximum is 10 or 11 mm/yr, and decreases slower at $>10/11 \text{ mm/yr}$ than it increases at $<10/11 \text{ mm/yr}$. Double-check interpretation.

Thank you for catching this mistake. At two locations in the paragraph, the optimum should be listed as 10 mm/yr instead of 8 mm/yr. This has been changed accordingly.

- Figure 4B: I’m curious why panel B doesn’t include a landscape line, which is helpful for interpreting net trends in 4A. Also it would be good to clarify how ecosystems are weighted (L393 says they are very different widths, so I assume different areas). % change may be a better way to apples-to-apples your inter-ecosystem comparisons (and more applicable across other systems where areal proportions differ widely from your model system).

The total landscape area for figure 4A remains constant throughout all simulations. The model is set up as a transect, so they have an inherent area of ecosystem width (m) by 1 (m). The idea to represent as % change in ecosystem size was excellent and is now included in the manuscript (Figure 4). Because we changed panel 4B to be percentages, it did not make sense to include a landscape line for this panel as the y axis now represents the percent change for a given ecosystem.

Figure 4. (a) Total landscape carbon stock, comprised of marsh, forest and bay ecosystems. (b) Shifts in landscape carbon stocks depend on the size of each component of the landscape, where change in ecosystem size is relative to the initial size of the ecosystem. Carbon stocks were calculated at the end of the model experiments (100 years) and are the sum of both biomass and soil carbon.

- Figure 4C: this is only referenced once that I find in the manuscript, and differs in structure and message from A and B. I suggest it should be a stand-alone supplemental figure, to make Figure 4 simpler and more cohesive.

Agreed, and changed accordingly.

- L234: extra space: "cycling , "

Changed

- L233: is increased accommodation space a result of eroding the marsh edge? Would be good to clarify that

No, the increased accommodation space in this case is driven by increased water levels in the bay. While the eroding marsh edge does add to the accommodation space, marsh edge erosion rates are similar across all SLR rates and don't lead to an increase in accommodation space with SLR. We clarified this in the text:

“Bay-bottom carbon stocks increase with SLR, driven by increased accommodation space from increased water depth (Fig. 4).”

- L337: this keeps the model much simpler, but, per comments above, some portion will likely be labile, and I think highlighting this assumption and the associated error in the main text is important to properly contextualize your results.

Agreed – see above response.

- L345: I’m curious about setting biomass to be maximized at an intermediate elevation. Is this based on vegetation assessment? If so, please cite, if not, I’m curious what impact that might have on the results.

A parabolic relationship between marsh elevation and biomass has been widely used across different marsh settings (i.e. tidal range, climate, plant community) (Mudd et al. 2004, van de Koppel et al. 2005, D’Alpaos et al. 2007, Kirwan and Murray 2007, Kirwan et al. 2010, Larsen and Harvey 2010, Lorenzo-Trueba et al. 2010, Mariotti and Carr, 2014, Kirwan et al. 2016, Carr et al. 2018) and comes from initial field experiments (Morris et al 2002 – reference number 50 in the manuscript). The conceptual idea behind a parabolic relationship is that at very low elevations, flooding is too frequent for marsh plants and biomass is low. At high elevations, flooding is not frequent enough, leading to low biomass. An alternate parameterization of marsh plant biomass is that suggested in Marani et al. 2013, which is not a parabola, but still has a maximum in plant biomass at intermediate elevations. Although we had cited the Morris et al. 2002 paper in the paper and methods, we added additional citations for models that have used this parameterization to make it clearer.

In the Model approach section, the sentence now reads and includes 3 citations:

“Belowground biomass, represented as a monoculture of *Spartina alterniflora*, is modeled as a parabolic function of elevation which peaks at an intermediate elevation ($B_{max} = 1000 \text{ gm}^{-2}$)^{31,36,38}.”

In the methods, the sentence now reads and includes 7 citations of the use of this parameterization:

“This quadratic relationship is most representative of *Spartina alterniflora*⁴⁸ and has been widely used in ecogeomorphic models of marsh evolution^{31,36,72–76}.”

References in Response:

Carr, Joel, et al. "Exploring the impacts of seagrass on coupled marsh-tidal flat morphodynamics." *Frontiers in Environmental Science* 6 (2018): 92.

Carr, J., G. Guntenspergen, and Matthew Kirwan. "Modeling marsh-forest boundary transgression in response to storms and sea-level rise." *Geophysical Research Letters* 47.17 (2020): e2020GL088998.

Kirwan, Matthew L., et al. "Sea level driven marsh expansion in a coupled model of marsh erosion and migration." *Geophysical Research Letters* 43.9 (2016): 4366-4373.

Kirwan ML, Guntenspergen GR, D'Alpaos A, Morris JT, Mudd SM, Temmerman S. 2010. Limits on the adaptability of coastal marshes to rising sea level. *Geophys Res Lett* 37:L23401. doi:10.1029/2010GL045489.

Kirwan ML, Murray AB. 2007. A coupled geomorphic and ecological model of tidal marsh evolution. *Proc Natl Acad Sci USA* 104:6118–22.

Lorenzo-Trueba J, Voller VR, Paola C, Twilley RR. 2010. Toward a model framework for sedimentary delta growth that accounts for biological processes. Abstract B33D-0427, AGU Fall Meeting, San Francisco.

Marani, Marco, Cristina Da Lio, and Andrea D'Alpaos. "Vegetation engineers marsh morphology through multiple competing stable states." *Proceedings of the National Academy of Sciences* 110.9 (2013): 3259-3263.

Mariotti, G., and J. Carr. "Dual role of salt marsh retreat: Long-term loss and short-term resilience." *Water Resources Research* 50.4 (2014): 2963-2974.